# Vitamin C-induced $CO_2$ capture enables high-rate ethylene production in $CO_2$ electroreduction

Jongyoun Kim[1,4], Taemin Lee [1,4], Hyun Dong Jung [2,4], Minkyoung Kim[1], Jungsu Eo[1], Byeongjae Kang[1], Hyeonwoo Jung[1], Jaehyoung Park[1], Daewon Bae[1], Yujin Lee[1], Sojung Park[3], Wooyul Kim[3], Seoin Back[2] ✉, Youngu Lee [1] ✉ & Dae-Hyun Nam [1] ✉

High-rate production of multicarbon chemicals via the electrochemical $CO_2$ reduction can be achieved by efficient $CO_2$ mass transport. A key challenge for C–C coupling in high-current-density $CO_2$ reduction is how to promote *CO formation and dimerization. Here, we report molecularly enhanced $CO_2$-to-*CO conversion and *CO dimerization for high-rate ethylene production. Nanoconfinement of ascorbic acid by graphene quantum dots enables immobilization and redox reversibility of ascorbic acid in heterogeneous electrocatalysts. Cu nanowire with ascorbic acid nanoconfined by graphene quantum dots (cAA-CuNW) demonstrates high-rate ethylene production with a Faradaic efficiency of 60.7% and a partial current density of 539 mA/cm², a 2.9-fold improvement over that of pristine CuNW. Furthermore, under low $CO_2$ ratio of 33%, cAA-CuNW still exhibits efficient ethylene production with a Faradaic efficiency of 41.8%. We find that cAA-CuNW increases *CO coverage and optimizes the *CO binding mode ensemble between atop and bridge for efficient C–C coupling. A mechanistic study reveals that ascorbic acid can facilitate *CO formation and dimerization by favorable electron and proton transfer with strong hydrogen bonding.

The electrochemical $CO_2$ reduction reaction ($CO_2$RR) to form value-added fuels and feedstocks is a promising route to achieve carbon neutrality and long-term energy storage[1]. The development of $CO_2$RR electrocatalysts has led to advances in selectivity for multicarbon ($C_{2+}$) chemicals such as ethylene ($C_2H_4$)[2–4] and ethanol ($C_2H_5OH$)[5,6] with high energy density and a high market price. However, it is a prerequisite to ensure profitability for the vitalization of $CO_2$RR-based chemical manufacturing. This requires an enhanced production rate for $C_{2+}$ chemicals[7]. These chemicals are formed by C–C coupling, which occurs via adsorbed carbon monoxide (*CO) dimerization on the surface of Cu active sites[8]. Although the product selectivity can be modulated by the design of active materials, including surface morphology, facet, defect, and alloy[9], the production rate of $CO_2$RR products, especially for $C_{2+}$ chemicals, is mainly affected by *CO formation on heterogeneous catalysts; the partial current density of $C_2H_4$ ($J_{C2H4}$) is proportional to the square of *CO coverage ($\theta_{CO}^2$)[10]. In general, at the potential for high current density, the hydrogen evolution reaction (HER) becomes more dominant than

[1]Department of Energy Science and Engineering, Daegu Gyeongbuk Institute of Science and Technology (DGIST), Daegu 42988, Republic of Korea.
[2]Department of Chemical and Biomolecular Engineering, Institute of Emergent Materials, Sogang University, Seoul 04107, Republic of Korea. [3]Department of Energy Engineering, Institute for Environmental and Climate Technology, Korea Institute of Energy Technology (KENTECH), Naju 58330 Jeollanam-do, Republic of Korea. [4]These authors contributed equally: Jongyoun Kim, Taemin Lee, Hyun Dong Jung. ✉e-mail: sback@sogang.ac.kr; youngulee@dgist.ac.kr; dhnam@dgist.ac.kr

CO2RR because of limited *CO coverage. Therefore, it has been a challenge to achieve CO2-to-*CO conversion in high current density CO2 electrolysis for high-rate C2+ chemical production.

CO2-to-*CO conversion is significantly hampered by the limited *CO2− formation, which is one of the main rate-determining steps (RDS) for the CO2RR[11]. To overcome this bottleneck, gas diffusion electrode (GDE)-based electrolyzers, such as flow cells and membrane-electrode-assembly (MEA), have emerged as an engineering approach to improve CO2 transport to catalysts. GDE enables the supply of a large amount of CO2 by providing gas flow directly to the electrocatalysts over the double- and triple-phase boundaries between gas CO2, liquid electrolytes, and solid catalysts[12,13]. In the GDE, microenvironment has been optimized in terms of CO2 flow rate[14], CO2 partial pressure[15], and catalyst architecture[8]. Also, molecular enhancement of CO2RR by combining molecular additives with heterogeneous catalysts has recently received great attention. They contribute to increasing *CO coverage by enhancing CO2 mass transport (increased local CO2 concentration) or optimizing hydrophobicity[2,3,16–24]. For instance, ionomers can provide CO2 transport channels that increase the local CO2 concentration near active sites and control the local pH and CO2/H2O ratio via selective ion conduction[7,17,18]. Furthermore, polymers in molecularly augmented GDEs have been introduced to increase the local CO2 concentration by optimizing porosity or hydrophobicity[20,25]. However, the strategy to promote proton-coupled electron transfer-based *CO formation from CO2 (g) has not yet been explored significantly for CO2RR with high productivity in heterogeneous electrocatalysts.

Here, we report molecularly enhanced CO2-to-*CO conversion and *CO dimerization for high-rate C2H4 production using ascorbic acid (AA). AA, also known as vitamin C, has been widely used as a reducing agent and antioxidant in nanomaterial synthesis and biochemical purposes[26,27]. When we store fruits to preserve AA, maintaining a CO2-deficient environment is essential because AA can react with CO2 and be oxidized to dehydroascorbic acid (DHA) with proton and electron donation[28]. Furthermore, AA has been utilized for CO2 capture in homogeneous catalysis approaches[29–31]. Inspired by this AA/DHA redox principle and CO2 capture property, we exploited AA as a promoter to capture CO2 near Cu, increase the *CO coverage and ensuing *CO dimerization on the surface of heterogeneous Cu catalysts.

To employ AA in heterogeneous catalysis with aqueous electrolytes, we pursued a strategy to immobilize water-soluble AA on electrocatalysts and achieve redox reversibility. We designed AA-augmented Cu nanowires (CuNWs) by applying graphene quantum dots (GQDs), which contain −OH and −COOH groups, as a mediator to anchor AA on the Cu surface with an ionomer. This nanoconfined AA on CuNW enhanced the CO2-to-*CO conversion during the CO2RR and resulted in high C2H4 productivity of heterogeneous Cu electrocatalysts. Unlike pristine CuNW (p-CuNW), which mainly produced C2H4 at low potential, CuNW with AA nanoconfined by GQDs (cAA-CuNW) boosted CO production over a similar potential range. As the potential increased for the high-current-density CO2RR, enriched CO formation in cAA-CuNW was dramatically converted to C2H4, while the main electrolysis product of p-CuNW was hydrogen (H2) because of limited CO2 mass transport. We found that this enables efficient CO2RR even in low CO2 concentrations, which can be extended to the CO2RR of flue gas. In situ Raman spectroscopy and operando X-ray absorption spectroscopy (XAS) studies enabled us to verify the effect of nanoconfined AA for inducing a high degree of *CO coverage and binding control between atop-bound CO (CO_atop) and bridge-bound CO (CO_bridge) on the reconstructed CuNW during the CO2RR. Grand canonical density functional theory (GC-DFT) revealed that the redox of AA/DHA enabled efficient electron/proton transfer to CO2 and multiple hydrogen bonding sites of AA, thereby improving CO2-to-*CO conversion and *CO dimerization on Cu.

## Results

### Fabrication of vitamin C-augmented catalysts

AA-augmented CuNW was leveraged for high-rate CO2-to-C2H4 conversion by nanoconfined AA on GQDs; favorable CO2-to-*CO conversion and *CO dimerization on the Cu surface (Fig. 1a). For CO2 capture, we harnessed the redox of AA/DHA; AA was oxidized and converted to DHA with electron and proton donation (Fig. 1b). A major bottleneck for the application of AA in heterogeneous catalysts is that AA is easily dissolved into the electrolyte due to its high solubility in aqueous solutions. Once it dissolves, it is difficult to reduce back to AA due to irreversible dehydration[32]. This stoichiometric consumption of AA degrades the sustainability of CO2RR systems. Additionally, reversible redox of AA/DHA is required for efficient CO2 capture. We improved the electroactive sustainability of AA by nanoconfined AA on GQDs. The confined AA ensures a redox-reversible environment through the improvement of the reduction reaction by DHA accumulation and stabilization in an aqueous electrolyte[32].

Nanoconfinement of AA was achieved by the reaction between AA and GQDs at 95 °C to reduce and functionalize the GQDs[33]. Note that excess reductant containing diverse oxygenated functional groups can remain on the reduced graphene surface and act as a multidentate hydrogen bonding donor[34,35]. The reduced GQDs can form 2-dimensional supramolecular systems that could effectively confine AA via physisorption including π interaction or hydrogen bonding. (Supplementary Fig. 1). Then, AA-nanoconfined GQDs were combined with CuNW through mild sonication, and cAA-CuNW uniformly maintained the high-aspect-ratio structure of CuNW (Supplementary Fig. 2).

We fabricated p-CuNW, CuNW with GQD (G-CuNW), CuNW with AA (AA-CuNW) and cAA-CuNW to understand the role of nanoconfined AA on GQDs in the CO2RR by comparing their catalytic reactions (Fig. 1c). Surface functionalization of CuNW was conducted by the interaction between the oxygen-containing functional groups of each reagent and the native oxide surface of the CuNW[36]. The surface structures of CuNWs were investigated by scanning electron spectroscopy (SEM). Compared to p-CuNWs, surface-modified CuNWs showed a rough surface because of the existence of AA and GQDs. However, they exhibited similar 1-dimensional (1D) structures even after surface modification (Supplementary Fig. 3).

The crystalline structures of CuNWs were investigated by transmission electron spectroscopy (TEM) (Fig. 1d–k, Supplementary Fig. 4). High-resolution (HR)-TEM of p-CuNWs revealed the presence of crystalline Cu (Fig. 1d, e). G-CuNW showed a rough surface morphology because of the polycrystalline GQD assembly (Fig. 1f). This indicates that GQDs uniformly cover the surface of CuNW. The lattice distance of the polycrystalline outer shell in G-CuNW was 0.24 nm, which corresponds to the (100) plane of graphene (Fig. 1g)[36,37]. AA-CuNW, composed of an amorphous shell with a thickness of ~3 nm, showed a more uniform surface than G-CuNW (Fig. 1h, i). The Cu surface of AA-CuNW was partially oxidized during surface functionalization because of the reaction with oxygen in an organic solvent (Supplementary Fig. 5). In the cAA-CuNW, both polycrystalline and amorphous regions coexisted, indicating that GQDs adequately confined AA on the surface of CuNWs (Fig. 1j, k). In the HR-TEM of CuNWs, lattice distance analysis and fast Fourier transform (FFT) confirmed the presence of pure Cu after surface functionalization (Supplementary Fig. 6).

### Chemical states of vitamin C-augmented catalysts

We investigated the chemical states of CuNWs after nanoconfinement to verify the presence of AA and GQD. In the TEM energy-dispersive spectroscopy (EDS) mapping, all CuNWs showed a uniform distribution of C and O along the Cu, implying that the surface of the CuNWs was coated by each reagent (Supplementary Fig. 7). In the EDS spectrum for Cu, C, and O, the atomic fraction of C in G-CuNW was 12.6%, while O was barely detected (Fig. 2a and Supplementary Fig. 8).

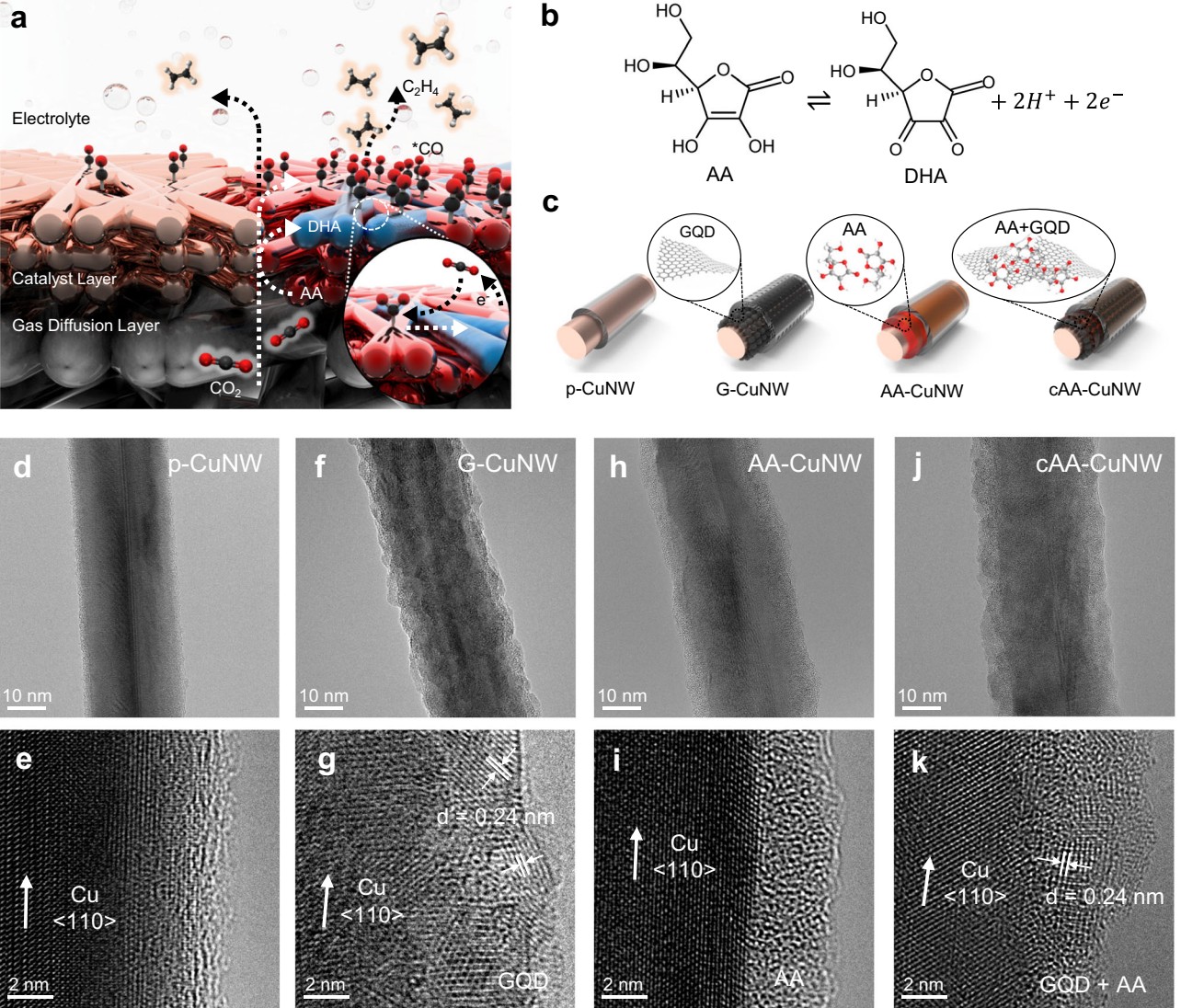

**Fig. 1 | CO₂ capture strategy and surface structures of AA-augmented CuNWs.**
**a** Schematic of enhanced CO₂-to-*CO conversion and *CO dimerization in cAA-CuNW for high-rate C₂H₄ production. **b** Redox of AA and DHA for CO₂ capture. **c** Schematic illustration of surface modification of CuNWs with GQD, AA, and nanoconfined AA on GQDs. An ionomer is coated on the outer surface of CuNWs during the fabrication of the GDE. TEM (top) and HR-TEM (bottom) images of (**d, e**) p-CuNW, (**f, g**) G-CuNW, (**h, i**) AA-CuNW, and (**j, k**) cAA-CuNW.

However, the fraction of O (15.8%) was higher than that of C (7.0%) in AA-CuNW. In cAA-CuNW, the fraction of O was lower than that of C (20.1%) but still showed a significant ratio of 12.8%, suggesting the coexistence of GQDs and AA.

The presence of AA in cAA-CuNW was verified by X-ray photoelectron spectroscopy (XPS). The C 1$s$ spectra of the G-CuNW showed a strong peak at ~284.5 eV, which corresponds to the $sp^2$ C group of the GQDs (Fig. 2b). In addition, peaks for C−O and O−C=O bonds were observed at ~285.8 and ~287.7 eV, respectively, originating from the oxygenated C functional group at the edge of the GQDs[30,37,38]. In the C 1$s$ spectra of cAA-CuNW, the intensity of the peaks for C−O and O−C=O bonds increased, indicating that cAA-CuNW contains more oxygenated C groups than G-CuNW (Fig. 2c). Since these peaks were not observed in the C 1$s$ spectra of GQDs separated from cAA-CuNW, the peaks for C−O and O−C=O bonds were attributed to the hydroxyl and carbonyl groups in AA (Supplementary Fig. 9).

In the Fourier transform infrared spectroscopy (FT-IR) analysis, we observed major peaks at approximately 1,666, 1,365, and 1,066 cm⁻¹, which corresponded to the C=C stretching vibration, C=C−O asymmetric stretching vibration of the enol-hydroxyl group, and C−O vibration in the functional groups of AA in AA-CuNW and cAA-CuNW (Fig. 2d)[39]. With TEM EDS, XPS, and FT-IR, comparison of thermogravimetric analysis (TGA) between G-CuNW and cAA-CuNW supported the presence of AA in cAA-CuNW (Supplementary Fig. 10). In the TGA of cAA-CuNW, an abrupt weight drop was observed at approximately 190 °C, consistent with the decomposition temperature of AA[40]. The X-ray diffraction (XRD) patterns of CuNWs exhibited a pure metallic Cu phase, indicating that the surface functionalization did not affect the chemical states of the active site in CuNWs (Fig. 2e). Consequently, it was inferred that AA is well confined on the surface of cAA-CuNW without any change in the chemical state of AA and Cu during surface modification.

We investigated the states of CuNWs after CO₂RR through XRD, SEM, TEM, and TEM EDS. XRD patterns of all CuNWs showed Cu₂O (111) peaks due to the oxidation of metal surface by electrolyte contact after CO₂RR (Supplementary Fig. 11). However, the 1D structures were maintained for all CuNWs as shown in SEM images (Supplementary Fig. 12), indicating that there was no significant structural transformation during CO₂RR. The crystal structure and atomic distribution of CuNWs were analyzed using TEM and TEM EDS (Supplementary

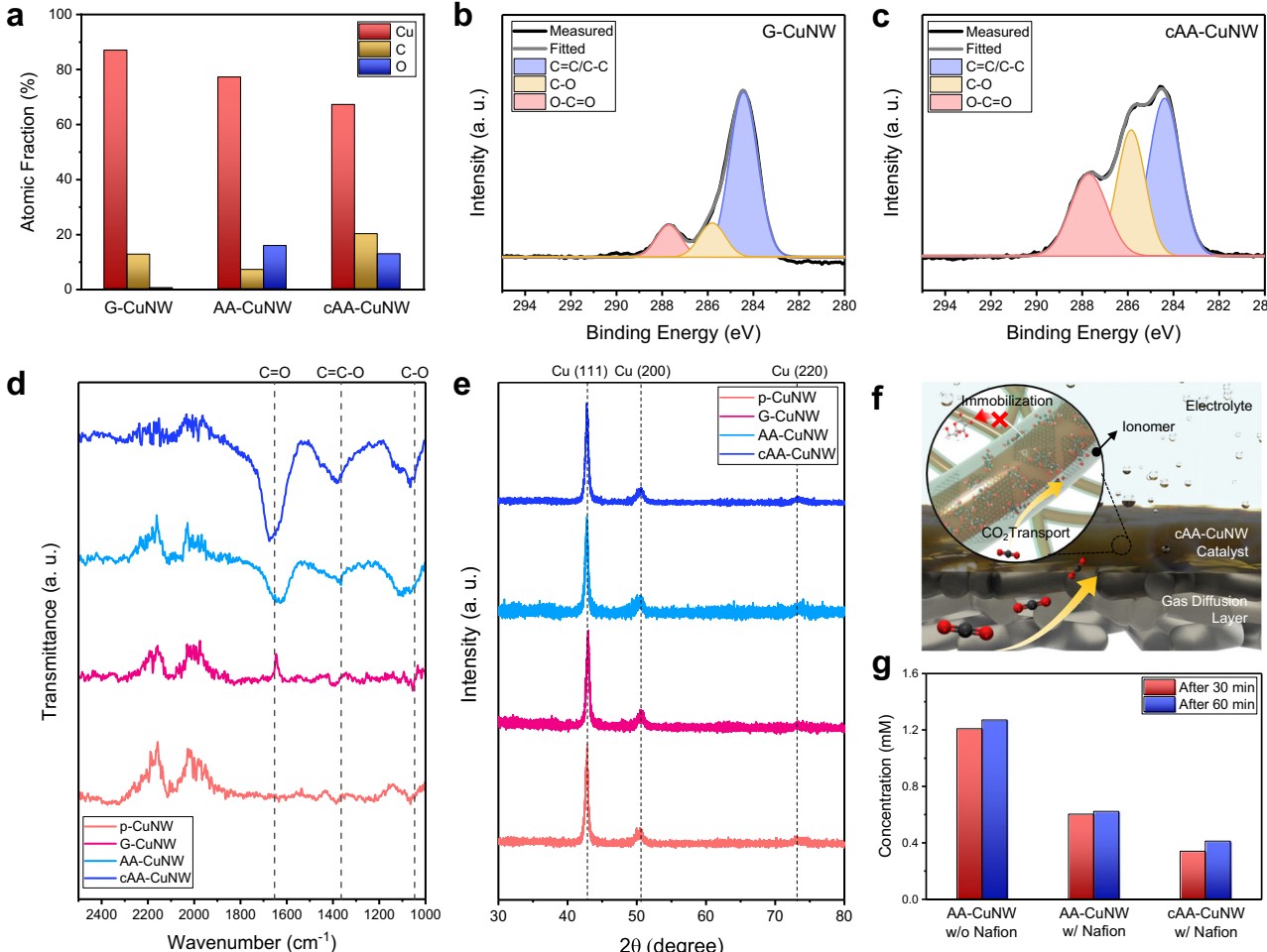

**Fig. 2 | Chemical states of AA-augmented CuNWs. a** Atomic fractions of Cu, C, and O for G-CuNW, AA-CuNW, and cAA-CuNW. C 1s XPS spectra of (**b**) G-CuNW and (**c**) cAA-CuNW. **d** FT-IR spectra of p-CuNW, G-CuNW, AA-CuNW, and cAA-CuNW. **e** XRD patterns of p-CuNW, G-CuNW, AA-CuNW, and cAA-CuNW. **f** Conceptual schematic of the surface components of Nafion ionomer-coated cAA-CuNWs for enhanced $CO_2$-to-*CO conversion and *CO dimerization during the $CO_2$RR. **g** Comparison of the amount of DHA extracted from AA-CuNW and cAA-CuNW with and without the Nafion ionomer coating. The extracted DHA concentration was analyzed by HPLC.

Fig. 13). CuNWs exhibited a rough $Cu_2O$ surface after catalysis, which corresponds to the XRD analysis results. In addition, the lattice of GQDs and amorphous nanostructure of AA was still observed at the outer shell of G-CuNW, AA-CuNW, and cAA-CuNW, suggesting that each material was located on the CuNW surface during $CO_2$RR. The elemental distribution in TEM EDS confirmed that the uniform distribution of Cu, C, and O atoms on the entire surface of the CuNW structure was maintained after $CO_2$RR (Supplementary Fig. 14).

The redox reversibility of nanoconfined AA on GQDs was verified by cyclic voltammetry (CV) measured in a 1 M KOH electrolyte without gas supply (Supplementary Fig. 15). CV of nanoconfined AA on GQDs at the surface of a glassy carbon electrode (GCE) showed paired oxidation and reduction peaks after repetitive scans. Early in the cycle, a strong oxidation peak was observed due to the existence of excess AA, and these unconfined AAs were oxidized by repetitive scans. On the other hand, the CVs of intact GCE and GQD-coated GCE showed no significant redox peaks in the same potential window. When redox molecules are confined in nanocavities and their collision frequency with the electrode surface increases, the electrochemical reversibility is improved due to the enhancement of electron transport[32]. Therefore, we determined that the redox reversibility of AA was enhanced by the nanoconfinement effect through GQDs, ensuring sustainable electron and proton transfer during the $CO_2$RR. Furthermore, we compared the CV of AA and nanoconfined AA on GQDs under $N_2$ and $CO_2$ gas (Supplementary Fig. 16). To minimize changes in CV curve due

to pH drop from bicarbonate formation, we proceeded CV with 0.1 M $KHCO_3$ electrolyte. Unlike AA, reversible AA/DHA redox is achieved in nanoconfined AA on GQDs under both $N_2$ and $CO_2$ gas. Therefore, we think that nanoconfined AA on GQDs can promote $CO_2$-to-*CO conversion by enhancing electron and proton transfer from reversible AA/DHA redox during $CO_2$RR (Supplementary Fig. 17).

To investigate the redox behavior of AA/DHA in the potential range of $CO_2$RR, we analyzed linear sweep voltammetry (LSV) of nanoconfined AA on GQDs before and after $CO_2$RR (Supplementary Fig. 18). We found that the most AAs were oxidized to DHA during LSV before $CO_2$RR. However, the current density of peak for the oxidation of AA significantly increased after 24 h $CO_2$RR at −1.8 V (vs RHE, non-iR corrected) (Supplementary Fig. 18b). This indicates that DHA was reduced to AA during $CO_2$RR, regenerating AA to continuously promote $CO_2$-to-*CO conversion in the potential range of $CO_2$RR.

An increased local $CO_2$ concentration and immobilization of AA on the CuNW surface could be achieved by a perfluorosulfonic acid (PFSA) ionomer used for GDE fabrication with catalysts (Fig. 2f). When Nafion was coated on the catalyst, hydrophilic functional groups ($-SO_3^-$) preferentially interacted with the metal surface and formed hydrophobic domain ($-CF_2$) channels through which electrolytes and gases could be transported[18]. The effect of Nafion on promoting the immobilization of AA was investigated by quantitative analysis of DHA in aqueous solution extracted from each catalyst. To evaluate the protective role of Nafion via the impermeability of AA and GQD, AA-

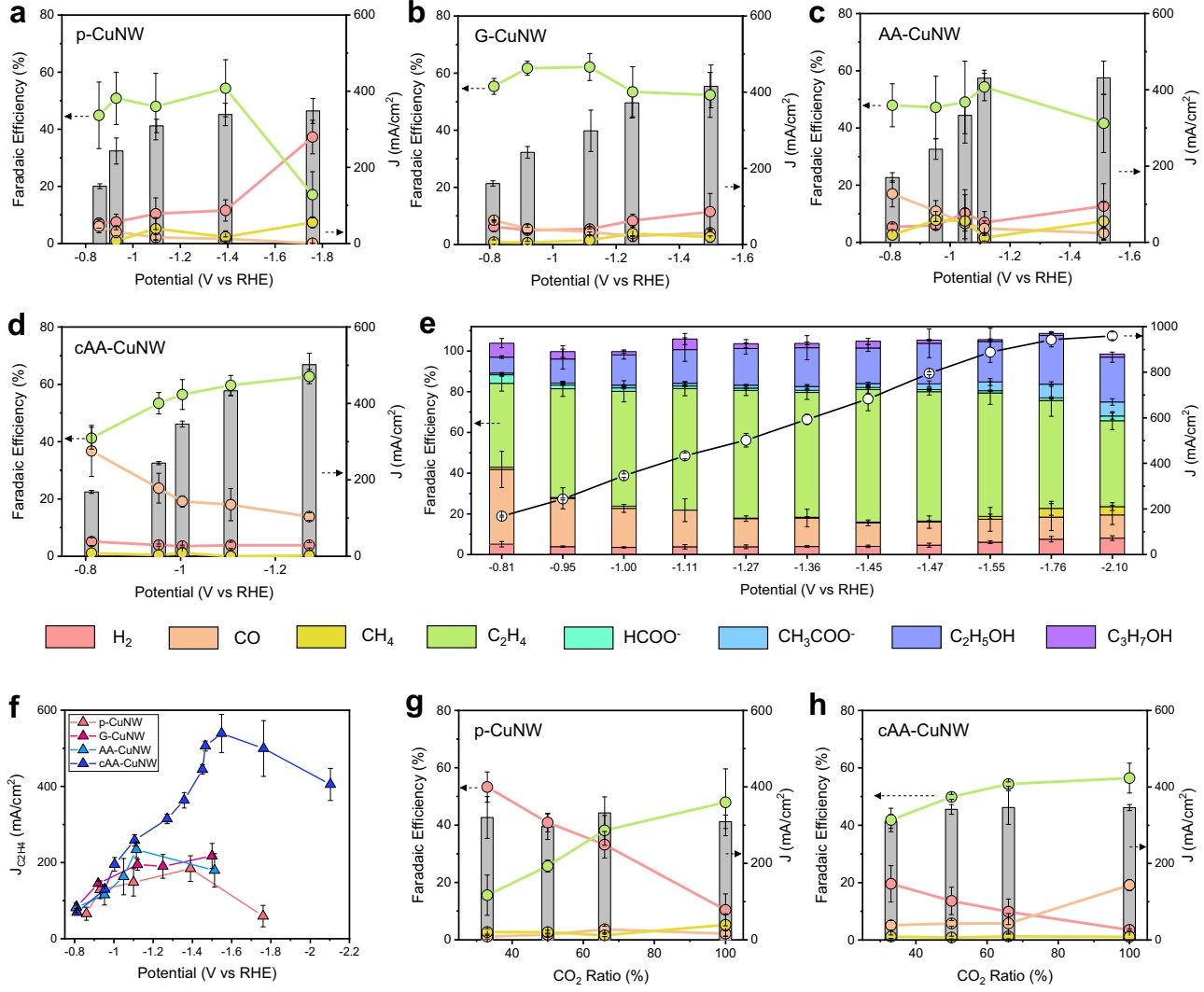

**Fig. 3 | High-rate C$_2$H$_4$ production of cAA-CuNW by enhanced CO$_2$-to-\*CO conversion and \*CO dimerization.** Gaseous product FEs and total current densities for (**a**) p-CuNW, (**b**) G-CuNW, (**c**) AA-CuNW, and (**d**) cAA-CuNW in the CO$_2$RR with 1 M KOH electrolyte. **e** Total product FEs and total current densities for the cAA-CuNW with applied potentials up to −2.10 V (vs RHE). **f** Comparison of $J_{C2H4}$ versus potentials of p-CuNW, G-CuNW, AA-CuNW, and cAA-CuNW. Gaseous product FEs and total current densities of (**g**) p-CuNW and (**h**) cAA-CuNW in the CO$_2$RR according to CO$_2$ ratios in CO$_2$ + Ar mixed gas. All the error bars represent standard deviation based on three independent samples.

CuNW and cAA-CuNW were prepared with and without Nafion coating. We quantified the extracted DHA by high-performance liquid chromatography (HPLC). DHA peaks were detected in the chromatographs of all catalysts at a retention time of ~5.8 min because dissolved AA was easily oxidized to form DHA during the extraction process (Supplementary Figs. 19 and 20)[32]. The concentration of DHA extracted from AA-CuNW without Nafion was 1.27 mM after 1 hour of extraction (Fig. 2g). However, the concentration of DHA dramatically decreased for AA-CuNW (0.62 mM) and cAA-CuNW (0.41 mM) with Nafion. We expect that the immobilization of AA was promoted by the laminar arrangement of Nafion on CuNW, which helps to prevent AA from penetrating into the aqueous electrolyte[41].

**Electrochemical CO$_2$RR of vitamin C-augmented catalysts**

The CO$_2$RR performances of the CuNWs were investigated in a flow cell electrolyzer with a 1 M KOH electrolyte (Fig. 3, Supplementary Figs. 22–25). In the p-CuNW, C$_2$H$_4$ was the main product at low potential ranges with a C$_2$H$_4$ Faradaic efficiency (FE) of 44.9% and CO FE of 6.1% at −0.86 V (Fig. 3a). However, when the potential exceeded −1.39 V (vs RHE) and reached −1.76 V (vs RHE), C$_2$H$_4$ FE decreased from

54.4 to 17.1%, and H$_2$ FE increased from 11.6 to 37.3% by limited CO$_2$ mass transport. G-CuNW showed more efficient C$_2$ product formation with a higher C$_2$H$_4$ FE of more than 50% than p-CuNW at a similar potential range (Fig. 3b). In addition, the maximum total current density of G-CuNW was 465 mA/cm$^2$ (Supplementary Fig. 25), higher than that of p-CuNW (348 mA/cm$^2$). The enhanced total current density of G-CuNW is attributed to the existence of GQDs. Graphitic C shells can stabilize Cu active materials by alleviating reconstruction at a reductive potential[4,42]. GQDs enhance the electrocatalytic activity, as the electron-donating functional group of GQDs promotes the activity of the CO$_2$RR by reducing the energy barrier of the RDS[43]. Indeed, intact GQDs participated in the CO$_2$RR (Supplementary Fig. 26). However, we think that hindered gas diffusion[44] and hydrophilic nature of GQDs do not show a promotional effect on CO$_2$ mass transport (Supplementary Fig. 27). As a result, G-CuNW exhibited enhanced C$_2$H$_4$ selectivity in the low potential range with a maximum C$_2$H$_4$ FE of 62.1% at −1.12 V (vs RHE), but H$_2$ FE increased over 10%, and C$_2$ selectivity decreased as the potential increased to −1.50 V (vs RHE).

When the AA was hybridized to CuNW without GQD, we observed increased CO production in the low-potential range, which clearly

contrasts with the low CO production in p-CuNW and CuNW hybridized with DHA (DHA-CuNW) (Supplementary Fig. 28). AA-CuNW showed a CO FE of 17.0% at −0.81 V (vs RHE), while p-CuNW and G-CuNW showed CO FEs lower than 10% in a similar potential range (Fig. 3c). However, this increased CO formation did not contribute to *CO dimerization for $C_2H_4$ production in the high current density $CO_2RR$. Although CO FE decreased to 3.2% at −1.51 V (vs RHE), the $H_2$ FE increased to 12.6%, and the $C_2H_4$ FE decreased to 41.6% (Supplementary Fig. 24c). This may have originated from the absence of a nanoconfinement effect, which helped to prevent the dehydration of the AA and induce reversible AA/DHA redox, even when the Nafion ionomer prevented the AA from being dissociated by the electrolyte.

cAA-CuNW exhibited dramatically elevated CO production with an FE of 36.7% at −0.81 V (vs RHE). As the potential increased to −1.27 V (vs RHE), the CO FE decreased from 36.7 to 13.8%, and the $C_2H_4$ FE increased from 41.2 to 62.7% (Fig. 3d). The enhanced CO production implied that a large amount of *CO was formed on the catalyst surface. Since the formation rate of the $C_2$ product was proportional to $\theta_{CO}^2$, the evolved CO formation rate in cAA-CuNW could contribute to enhancing $CO_2$-to-*CO conversion for high-rate $C_2H_4$ formation. In particular, the ratio of $C_2H_4$ FE to methane ($CH_4$) FE ($C_2H_4$ FE/$CH_4$ FE) in cAA-CuNW is much higher than that of other CuNWs (Supplementary Fig. 29). This indicates that *CO dimerization is promoted in cAA-CuNW. In addition, cAA-CuNW showed a significantly low HER with an $H_2$ FE of 3.8% at −1.27 V (vs RHE). In cAA-CuNW, highly efficient $C_2H_4$ production was well maintained even when the potential reached −2.1 V (vs RHE) with a total current density of 960 mA/$cm^2$ (Fig. 3e).

When we compared partial current densities for $H_2$, CO, and $C_2H_4$, cAA-CuNW exhibited the highest $C_2H_4$ selectivity and productivity compared to other CuNWs; maximum $J_{C2H4}$ of 539 mA/$cm^2$ with $C_2H_4$ FE of 60.7% at −1.55 V (vs RHE), which is 2.9-fold higher than the highest $J_{C2H4}$ of p-CuNW with 184 mA/$cm^2$ at −1.39 V (vs RHE) (Fig. 3f and Supplementary Fig. 30). This is one of the highest $J_{C2H4}$ values among previously reported Cu-based $CO_2RR$ electrocatalysts (Supplementary Table 1). cAA-CuNW also showed the highest selectivity and productivity of CO and $C_2H_4$ in the electrochemically active surface area (ECSA)-normalized activity (Supplementary Figs. 31 and 32). The highest $J_{C2H4}$ of cAA-CuNW is confirmed after considering the effect of surface roughness. This reveals that nanoconfined AA on GQDs promotes the intrinsic $CO_2RR$ activity of cAA-CuNWs. We found that the overpotential at maximum $J_{C2H4}$ can be lowered by increasing the concentration of electrolyte such as $J_{C2H4}$ of 453 mA/$cm^2$, $C_2H_4$ FE of 56.3% at −0.57 V (vs RHE) in 2 M KOH electrolyte (Supplementary Fig. 33). However, when GQD and AA mixtures were deposited on the surface of CuNWs without any preceding reaction, there was no trend toward the enhancement of the production of CO and $C_2H_4$ (Supplementary Fig. 34). This revealed that the interaction between AA and GQD is key for the nanoconfinement effect, which can promote $CO_2$-to-*CO conversion and *CO dimerization by reversible AA/DHA redox and nanoconfined AA on GQDs.

Furthermore, we investigated $CO_2RR$ of p-CuNW and cAA-CuNW in low $CO_2$ concentration by controlling $CO_2$ ratio in $CO_2$ + Ar mixed gas to further prove the effect of nanoconfined AA on $CO_2$-to-*CO conversion (Fig. 3g, h and Supplementary Fig. 35). As the $CO_2$ ratio decreased in the mixed gas, p-CuNW showed a dramatic increase of $H_2$ selectivity and decrease of $C_2H_4$ selectivity ($H_2$ FE of 53.3% and $C_2H_4$ FE of 15.6% at the $CO_2$ ratio of 33%). In contrast, cAA-CuNW exhibited $H_2$ FE of 19.6% and $C_2H_4$ FE of 41.8% even at the $CO_2$ ratio of 33%. Efficient $C_2H_4$ production of cAA-CuNW was maintained by promoted $CO_2$-to-*CO conversion even in low $CO_2$ concentration.

To confirm the stability of nanoconfined AA on GQDs in promoting $CO_2RR$, long-term $CO_2RR$ was conducted in a flow cell electrolyzer with a 1 M KOH electrolyte (Supplementary Fig. 36). $C_2H_4$ selectivities of p-CuNW and cAA-CuNW were compared according to react ion times at a total current density of 300 mA/$cm^2$. $C_2H_4$ FE for

cAA-CuNW was maintained over 50% for 8 h, while that of p-CuNW abruptly decreased to 21.3% within 2 h. Also, similar FT-IR spectra of the GDE before and after the $CO_2RR$ stability test confirmed that the nanoconfined AA on GQDs was stable in cAA-CuNW during $CO_2RR$ (Supplementary Fig. 37). The $CO_2RR$ stability of cAA-CuNW was further investigated in a zero-gap membrane electrode assembly (MEA) electrolyzer with 0.1 M $KHCO_3$ anolyte (Supplementary Fig. 38). cAA-CuNW exhibited outstanding stability of $C_2H_4$ production for 168 h at a total current density of 150 mA/$cm^2$. These results indicate that the nanoconfined AA is stably immobilized in cAA-CuNW and continuously promote $CO_2$-to-*CO conversion and *CO dimerization to enhance $CO_2RR$ productivity.

## Real-time analysis to track $CO_2RR$ intermediates and Cu reconstruction

To study the origin of enhanced $C_2H_4$ productivity in cAA-CuNW, in situ Raman spectroscopy analysis was performed during the $CO_2RR$ with a 1 M KOH electrolyte (Fig. 4, Supplementary Fig. 39). This enabled tracking of the interaction of *CO intermediates with the catalyst surface, as well as the *CO binding mode for C−C coupling. Specific Raman peaks were observed for *CO in the regions of 200–300, 300–400, 1950–2000, and 2050–2100 $cm^{-1}$, which were related to Cu−CO rotation, Cu−CO stretching, $CO_{bridge}$, and $CO_{atop}$, respectively[45,46]. The other peaks observed at 525–534 and 615–630 $cm^{-1}$ of OCP were related to the interactions of the natural oxides with the Cu surface in CuNWs[47].

Securing the *CO intermediate on the active materials is important to maintain $C_2$ selectivity in the high current density $CO_2RR$. The surface *CO coverage of CuNWs was investigated by analyzing the Cu−CO binding peaks at 200–400 $cm^{-1}$. The origin of the Cu-CO binding peaks from $CO_2$ gas was confirmed through the comparison with in situ Raman spectroscopy under $N_2$ gas (Supplementary Fig. 40). cAA-CuNW, where nanoconfined AA on GQDs exists on the Cu surface, presented a significantly higher peak intensity for Cu−CO rotation and Cu−CO stretching than other CuNWs such as p-CuNW, G-CuNW, and AA-CuNW (Fig. 4a, c, e, g, and Supplementary Fig. 41). However, the peak intensities of p-CuNW were the lowest, which corresponds to the rapid increase of HER in the low-potential range[48]. Considering that cAA-CuNW exhibited a lower $*CO_2^-$ peak intensity at 1500–1600 $cm^{-1}$ compared to p-CuNW (Supplementary Fig. 42)[49], high *CO coverage of cAA-CuNW is attributed to the promoted $CO_2$-to-*CO conversion. To further investigate the effect of AA on *CO dimerization, we analyzed the Raman peaks for $CO_{bridge}$ and $CO_{atop}$ at ~1950–2100 $cm^{-1}$, representing C≡O stretching of the adsorbed CO on the metal surface. The ratio between $CO_{atop}$ and $CO_{bridge}$ was strongly related to $C_2$ product selectivity[2,50]. In general, the C − C coupling energy barrier decreased in the order of $CO_{bridge}$ to $CO_{bridge}$ > $CO_{atop}$ to $CO_{atop}$ > $CO_{atop}$ to $CO_{bridge}$. Therefore, maintaining an optimal $CO_{bridge}$/$CO_{atop}$ ratio is essential for efficient $C_{2+}$ chemical production. In the C≡O stretching region, $CO_{atop}$ and $CO_{bridge}$ were simultaneously formed in AA-CuNW and cAA-CuNW at all $CO_2RR$ potentials ranging from −0.4 to −0.8 V (vs RHE, non-iR corrected), whereas the $CO_{bridge}$ peak almost disappeared as the reductive potential increased in p-CuNW (Fig. 4b, d, f, h, and Supplementary Fig. 41). We found that AA-CuNW and cAA-CuNW maintain adequate $CO_{bridge}$/$CO_{atop}$ ratio even when the potential increases up to −0.8 V (vs RHE, non-iR corrected), while p-CuNW and G-CuNW exhibit excessive $CO_{atop}$ or $CO_{bridge}$ as potential increases (Fig. 4i). Note that strong Cu−CO binding peaks and optimal $CO_{bridge}$/$CO_{atop}$ ratio in in situ Raman spectroscopy of cAA-CuNW were similarly observed under $CO_2$ + Ar mixed gas (Supplementary Fig. 43). Therefore, cAA-CuNW with C−C coupling between the $CO_{bridge}$ and $CO_{atop}$ ensemble facilitates *CO dimerization for $C_{2+}$ chemical formation.

We investigated the reconstruction of Cu active materials in p-CuNW and cAA-CuNW by operando X-ray absorption spectroscopy (XAS) with a flow-cell-type reactor (Fig. 4j, k). We analyzed the

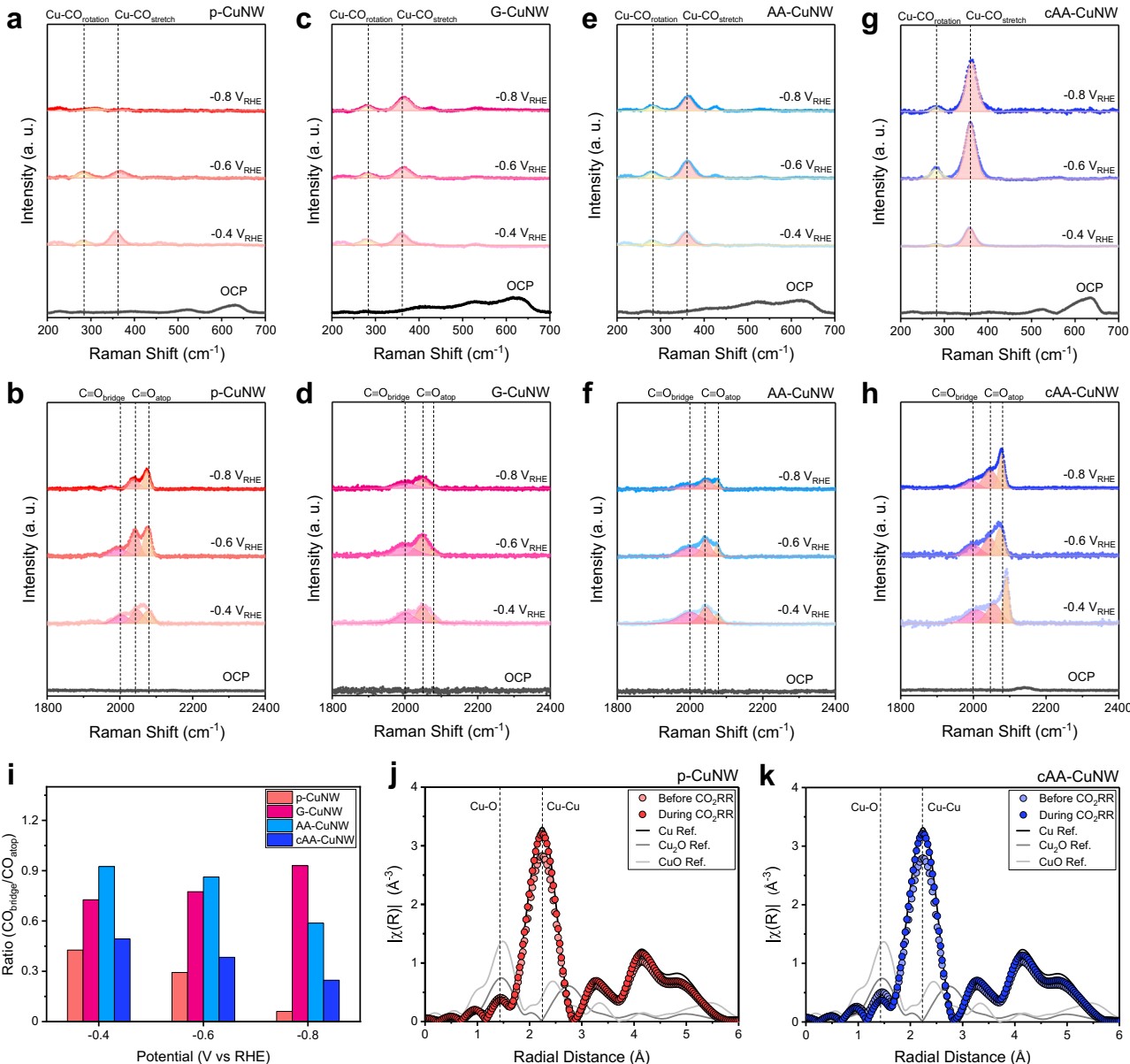

**Fig. 4 | Real-time analysis of CuNWs with different degrees of surface hybridization during the CO₂RR.** In situ Raman spectra of (**a, b**) p-CuNW, (**c, d**) G-CuNW, (**e, f**) AA-CuNW, and (**g, h**) cAA-CuNW obtained during CO₂RR according to the applied potentials in the region of 200-700 cm⁻¹ (top) and 1,800-2,400 cm⁻¹ (bottom). **i** Comparison of integral area ratios of CO$_{bridge}$ and CO$_{atop}$ for CuNWs. Operando EXAFS spectra of (**j**) p-CuNW and (**k**) cAA-CuNW during the CO₂RR.

oxidation states and coordination number (CN) of Cu during the CO₂RR of CuNWs at a cathodic potential of −1.4 V (vs RHE, non-iR corrected) in a 1 M KOH electrolyte. The Cu K-edge X-ray absorption near edge structure (XANES) spectra revealed that both p-CuNW and cAA-CuNW have the oxidation state of metallic Cu⁰, which was maintained during the CO₂RR (Supplementary Fig. 44). The CNs of p-CuNW and cAA-CuNW were investigated by extended X-ray absorption fine structure (EXAFS) spectroscopy (Fig. 4j, k and Supplementary Fig. 45). In the p-CuNW, the Cu−Cu bonding CN increased from 10.269 to 11.557 during the CO₂RR, indicating the reconstruction of Cu by reductive potential (Supplementary Table 2). We found that the Cu−Cu bonding CN of cAA-CuNW increased from 10.266 to 11.706, similar to that of p-CuNW. This similar Cu reconstruction between p-CuNW and cAA-CuNW revealed that nanoconfinement of AA on GQDs did not affect the Cu reconstruction behavior and that enhanced CO₂-to-*CO conversion and *CO dimerization originated from the augmentation of AA on CuNWs, not active materials.

## Mechanistic study on the role of vitamin C in the CO₂RR

To understand the effect of AA on improved C$_{2+}$ production, grand canonical density functional theory (GC-DFT) calculations were performed. We focused on the key steps of the electrochemical CO₂RR and compared the energetics and kinetics of Cu (100) with AA-decorated Cu (100), referred to as AA/Cu (100). The main steps include (1) CO₂ adsorption, (2) *COOH formation, (3) *CO formation, and (4) *CO dimerization. The competitive HER was additionally investigated. Note that easier *CO formation would increase *CO coverage on the surface, and more facile *CO dimerization would improve the selectivity of C$_{2+}$ products.

We present the Gibbs free energy diagrams of each step at pH 14 and 0 V (vs RHE) (Fig. 5a). Both Cu (100) and AA/Cu (100) similarly preferred CO₂ chemisorption. One C-O bond was oriented parallel to the Cu surface (C-O$_s$) and the other C-O bond was bent toward the solvent (C-O$_e$), where O$_s$ and O$_e$ stand for oxygen atoms near the surface and the electrolyte, respectively (Supplementary Fig. 47). The

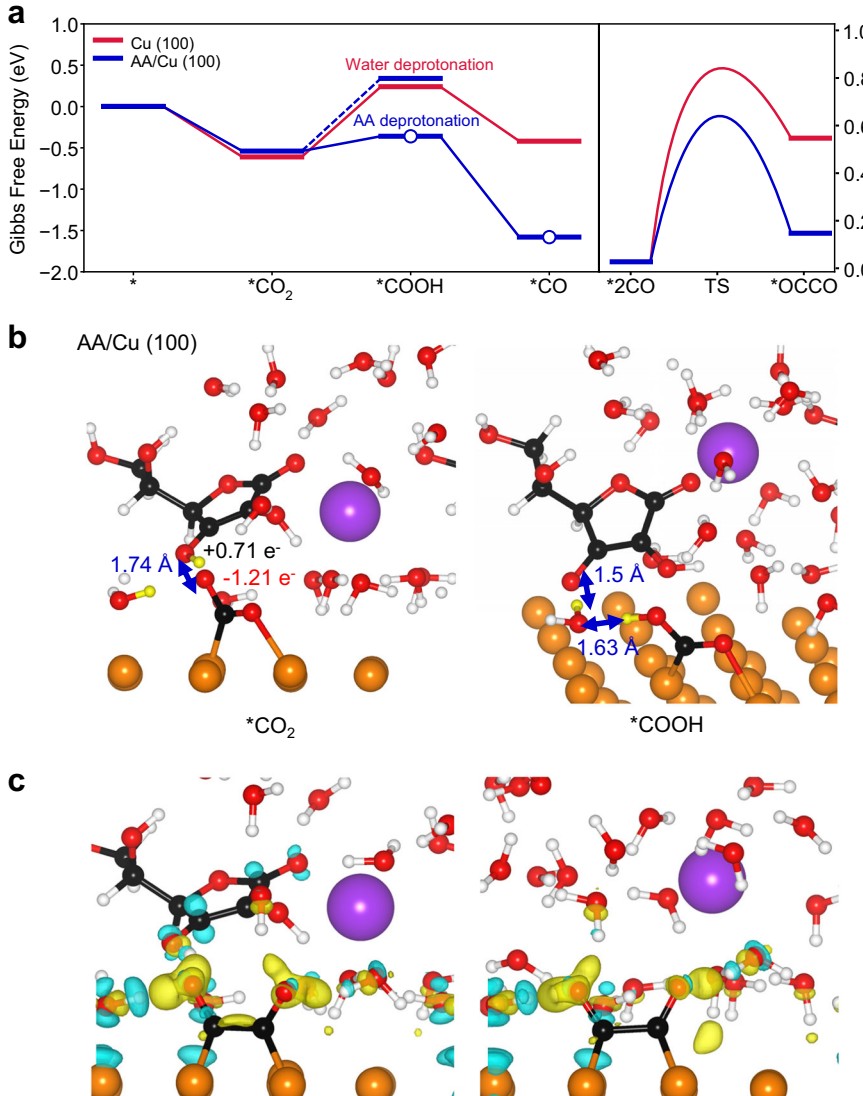

**Fig. 5 | Computational modeling of the CO₂RR on Cu (100) and AA/Cu (100).**
**a** Gibbs free energy diagram of (left) CO₂-to-*CO conversion and (right) *CO
dimerization at 0 V (vs RHE). White circles indicate the reaction pathway involving
the deprotonation of AA. H₂O is the proton source otherwise. **b** The atomic
structure of (left) the initial and (right) the final states of *CO₂ protonation from AA
on AA/Cu (100) (*CO₂ + H₂O + AA + e⁻ → *COOH + H₂O + ASC⁻). **c** Charge density
difference (△ρ) of *OCCO adsorption on (left) AA/Cu (100) and (right) Cu (100).

Color code: black (C), white (H), yellow (H), red (O), purple (K), and orange (Cu).
Yellow H atoms are considered for the protonation during the CO₂RR. The yellow
and blue area represent an electron accumulation and depletion with an isosurface
level of 0.005 e/Å³, respectively. The charge density difference is calculated as
$\rho_{total} - \rho_{ads+surf.} - \rho_{solv.}$, where $\rho_{total}$, $\rho_{ads+surf.}$ and $\rho_{solv.}$ correspond to charge
densities of the total system, the catalyst surface with adsorbates and solvent lay-
ers, respectively.

observed anisotropic configuration of the CO₂ adsorption is in con-
sistent with literature[51]. The Bader charge analysis demonstrates that
*CO₂ is highly negatively charged in both cases (−1.16 e⁻ on Cu (100)
and −1.07 e⁻ on AA/Cu (100)), indicating a strong CO₂ adsorption due
to the electron transfer (Supplementary Table 3)[52].

The adsorbed *CO₂ is then protonated to form *COOH. On Cu
(100), we considered a water molecule as the protonation source, and
the reaction Gibbs energy ($\Delta G_{*CO_2 \to *COOH}$) was found to be endother-
mic (0.86 eV). On AA/Cu (100), we evaluated both the water molecule
and AA as potential protonation sources (R2b for water and R2c for
AA)[53]. While the protonation from water exhibited a similar endo-
thermic reaction Gibbs energy of 0.88 eV, protonation from AA proved
to be thermodynamically more favorable, with a significantly lower
reaction Gibbs energy ($\Delta G_{*CO_2 \to *COOH} = 0.18$ eV). This enhanced favor-
ability can be attributed to lower pKa value of AA (4.04)[54] compared to
water (14), leading to a lower electrochemical deprotonation barrier[55].

To further elucidate the origin of the enhanced energetics, we
inspected the adsorption configurations before and after the proto-
nation. Before the protonation, on Cu (100), the hydrogen in the
proton source was positively charged by +0.67 e⁻ and the hydrogen
bonding distance ($d_{O-H}$) between $O_e$ was 1.9 Å (Supplementary Fig. 48).
In contrast, on AA/Cu (100), $H_{OX1}$ exhibited a more positively charged
state (+0.71 e⁻) and a shorter $d_{O-H}$ of 1.74 Å (Fig. 5b). After protonation,
the hydrogen bonding network within the solid-electrolyte interface
underwent significant reconstruction in the case of AA/Cu (100). The
distance between O in water and H in *COOH was 1.63 Å, and the
distance between H in water and O in ASC⁻ was 1.5 Å. However, on Cu
(100), we did not observe a significant reconstruction following the
protonation of *CO₂ from water. As a result, *COOH and ASC⁻ ion
formed a more robust hydrogen bond network with water, thereby
stabilizing the system. This observation suggests that the presence of
AA promotes the favorable protonation of *CO₂ to *COOH. Further

protonation of *COOH to form *CO was energetically favorable in both cases, resulting in higher *CO coverages on AA/Cu.

We subsequently investigated the dimerization of *2CO, a key determinant of the selectivity of $C_{2+}$ products such as $C_2H_4$[56]. The dimerization of *2CO on Cu (100) was found to be endothermic, exhibiting the reaction energy of 0.52 eV and the activation energy of 0.81 eV. The introduction of AA on the surface decreased these energies to 0.13 eV and 0.62 eV, respectively, demonstrating that AA facilitates the dimerization. The charge density difference plot and the Bader charge analysis confirmed more significant electron transfer to *OCCO on AA/Cu (100) ($-1.53\,e^-$ and $+0.18\,e^-$ for adsorbate and Cu surface, respectively) compared to Cu (100) ($-1.39\,e^-$ and $+0.08\,e^-$), making the interaction between *OCCO and the surface stronger through a more favorable electrostatic interaction (Fig. 5c)[57,58].

We also examined the competitive HER. The energetics of the first protonation step to form the adsorbed *H were found to be less favorable compared to $*CO_2$ adsorption, both with and without AA. Although the introduction of AA lowered the energy barrier of *H formation, the second protonation step remained unfavorable compared to $*CO_2$ adsorption (Supplementary Fig. 49). To sum up, GC-DFT calculations confirmed that the introduction of AA on Cu catalyst surface facilitated the $CO_2RR$, increasing *CO coverage on the surface and lowering the activation barrier of the rate-determining *CO dimerization, leading to higher production of $C_{2+}$ products.

## Discussion

We report vitamin C-induced $CO_2$ capture for effective $CO_2$-to-$C_2H_4$ conversion. AA was introduced to CuNWs for high rate $C_2H_4$ production by promoting electron and proton transfer and strong hydrogen bonding. By leveraging GQDs as a mediator to anchor water-soluble AA on CuNWs, we immobilized AA with an ionomer and enhanced the redox reversibility of AA, enabling sustainable $CO_2$ capture of AA for high-current density $CO_2RR$. This nanoconfined AA on Cu can steer the pathway toward $C_2H_4$ by securing $CO_2$-to-*CO conversion and *CO dimerization at high current density $CO_2RR$. cAA-CuNW exhibited higher CO selectivity than p-CuNW at a similar range of low electrolysis potential, indicating that nanoconfined AA effectively increased *CO coverage during the $CO_2RR$. cAA-CuNW demonstrated $CO_2RR$ with a $C_2H_4$ FE of 60.7% and $J_{C2H4}$ of 539 mA/cm$^2$ in 1 M KOH, 2.9-fold higher than the $J_{C2H4}$ of p-CuNW. In the $CO_2RR$ under $CO_2$ + Ar mixed gas ($CO_2$ ratio of 33%), cAA-CuNW exhibited $C_2H_4$ FE of 41.8% and $H_2$ FE of 19.6%, while p-CuNW exhibited $C_2H_4$ FE of 15.6% and $H_2$ FE of 53.3%. Efficient $CO_2RR$ of cAA-CuNW at low $CO_2$ concentration confirms the promoted $CO_2$-to-*CO conversion, applicable to the $CO_2RR$ of flue gas. In situ Raman spectroscopy and operando XAS revealed that enhanced *CO coverage and a judiciously controlled $CO_{bridge}/CO_{atop}$ ratio for efficient C−C coupling was induced by the augmented AA on CuNWs. In GC-DFT, the Gibbs free energy diagram reveals that AA can facilitate *CO formation and dimerization by promoting electron/proton transfer and strong hydrogen bonding on the CuNW surface. This strategy can simultaneously contribute to optimizing *CO coverage for mass production of $C_{2+}$ chemicals by combining with other molecular strategies for enhancing $CO_2$ mass transport in GDE. We believe that vitamin C-promoted $CO_2$ conversion, enabled by leveraging carbon mediators for bridging homogeneous and heterogeneous catalysis, can provide an avenue for compelling high-rate $C_{2+}$ chemical manufacturing.

## Methods

### Nanoconfined AA on GQD preparation
GQDs (Sigma–Aldrich, 0.1 mg/mL) and L(+)-ascorbic acid (ACROS, 99%, 6 mM) were dissolved in 50 mL of D.I. water. Ammonia solution (Daejung Chemicals & Metals Co., 25%, 0.1 mL) was added to ensure the colloidal stability of the GQDs. The mixtures reacted at 95 °C for 1 h. Then, the solution was evaporated and redispersed in 5 mL of D.I. water.

### Synthesis of the CuNWs and surface functionalization
CuNWs were prepared by hydrothermal methods[37]. Anhydrous copper chloride ($CuCl_2$, Alfa Aesar, 13 mM), D-glucose (Sigma–Aldrich, 11 mM), and hexadexylamine (TCI, 56 mM) were dissolved in D.I. water (280 mL). The mixtures reacted in a hydrothermal reactor at 120 °C for 24 h and centrifuged with D.I. water. The product was then separated using hexane to collect CuNWs and dispersed in isopropyl alcohol (IPA). To prepare surface-functionalized CuNWs, 0.5 mL of GQDs, AA (5 mg/mL), or nanoconfined AA on GQDs was injected in methanol (20 mL). After adding CuNW solution (3 mg/mL, 2 mL), surface functionalization was conducted with mild sonication for 20 min. After that, the solution was washed with IPA by centrifuging at 2490 $g$ for 15 min and stored in IPA.

### Characterization of the CuNWs
The nanostructures of the catalysts were confirmed by an FEI Titan TEM (THEMIS Z, Thermo Fisher Scientific) operated at 300 kV. TEM sampling was performed by drop casting the catalysts on lacey formvar carbon-coated square Au grid. The crystal structures and chemical composition of the catalysts were analyzed by XRD (Miniflex 600 Mini, Horiba) and XPS (ESCALAB 250Xi, Thermo Fisher Scientific). XPS sampling was performed by drop casting of the catalyst on P type Boron doped Si wafer. For the characterizations of CuNWs after $CO_2RR$ (XRD, SEM, TEM, TEM EDS), each electrode was operated in a flow cell elctrolyzer at the applied potential of $-3.2\,V$ (vs Ag/AgCl) for 1 h (Supplementary Figs. 11–14). The redox reversibility of nanoconfined AA on GQDs was confirmed by an electrochemical analyzer (VSP, Bio-Logic). The electrode was prepared by coating 30 μL of nanoconfined AA on GQDs on a GCE with a radius of 3 mm. Then, 10 μL of Nafion was subsequently coated on the electrode to prevent dissolution into the electrolyte. The CV measurement was conducted using a three-electrode cell consisting of a Pt wire (counter electrode) and Ag/AgCl (reference electrode) with a scan rate of 50 mV/s. The chemical analysis of DHA after elution from the catalysts was performed by HPLC (Vanquish Core HPLC, Thermo Scientific). After sufficient chemical extraction by stirring the film in D.I. water, the extracted DHA was analyzed using an Aminex HPX-87H cation exchange column with a mobile phase of 0.5 mM sulfuric acid aqueous solution. The column temperature was maintained at 25 °C, and the flow rate was set at 0.6 mL/min. The chromatographic peaks were detected by a UV detector at a wavelength of 200 nm. The intermediates of the catalysts during the $CO_2RR$ were analyzed by in situ Raman spectroscopy (XploRA$^{TM}$ PLUS Raman spectrometer, HORIBA). An altered flow cell with a water immersion objective lens (60×) and 785 nm laser were used. The $CO_2$ gas flow was fixed at 50 sccm by a mass flow controller (MFC). In situ Raman spectroscopy was measured up to the potentials of $-0.8\,V$ (vs RHE, non-iR corrected) due to laser focusing interference induced by gaseous products. The data were collected with an acquisition time of 10 s and acquired ten times in all measurements. Real-time analysis of the chemical states and bonding nature was performed by means of operando XANES and EXAFS. The XAS measurements were conducted at the 10 C X-ray beamline of Pohang Accelerator Laboratory (Pohang, Republic of Korea).

### Electrode preparation and electrochemical $CO_2RR$ measurement
Fifteen milligrams of CuNW were prepared as dispersed in IPA. Sixty microliters of Nafion ionomer (Sigma–Aldrich, 5 wt%) was mixed to increase both the immobilization of the catalysts and the $CO_2$ transport capacity. After vortex mixing, the solution was spray-coated on a porous polytetrafluoroethylene substrate (PTFE, Sterlitech) with an airbrush gun using $N_2$ gas to fabricate the GDE. To ensure uniform layer formation, spray coating was conducted on a 60 °C hot plate.

For the $CO_2RR$ experiment, Ag/AgCl was used as the reference electrode, and Ni foam was used as the counter electrode. The anolyte

and catholyte were separated by the anion exchange membrane. The electrocatalytic $CO_2RR$ performances of CuNWs were investigated at different cathodic potentials in a flow cell electrolyzer with a 1 M KOH electrolyte. The potential of Ag/AgCl was converted to RHE and the RHE conversion equation is as follows:

$$E_{RHE} = E_{Ag/AgCl} + 0.197(E^0) + 0.059*pH \qquad (1)$$

We performed the calculation of 80% iR compensation losses between the Ag/AgCl and working electrode through electrochemical impedance spectroscopy (EIS) analysis.

The gas products of $CO_2RR$ were analyzed by gas chromatography (GC, Micro GC Fusion, INFICON Inc.) equipped with two thermal conductivity detectors (TCDs) employing different carrier gases (argon and helium). The liquid products were analyzed by nuclear magnetic resonance spectrometry (NMR, Bruker). The mixture of catholyte, $D_2O$ and dimethyl sulfoxide as an internal standard was collected in NMR tubes. The Faradaic efficiency of each product was calculated by the following equation:

$$FE(\%) = \frac{z \cdot n \cdot F}{Q} \qquad (2)$$

where $z$ and $n$ are the number of electrons exchanged and moles of products, respectively; $F$ is the Faradaic constant; and $Q$ is the input charge. Gas product GC data per each sample were collected as the average value measured at 5, 30, and 55 min intervals. The error bars for $CO_2RR$ data (gas and liquid products) represent standard deviation based on three independent samples. In the $CO_2RR$ under $CO_2 + Ar$ mixed gas, all measurements were conducted at the applied potential of −2.2 V (vs RHE, non-iR corrected) with 1 M KOH electrolyte. Gas flow rate was fixed to 60 sccm.

### Grand canonical density functional theory (GC-DFT) calculations

The conventional calculation approach based on computational hydrogen electrode (CHE) method[59] simulates all systems at constant charge. Although this method is advantageous in understanding electrochemical processes with only a few DFT calculations[60,61], it could mislead the fundamental misunderstanding since the actual processes take place at a constant potential. This is because the systems described with the CHE method remain neutral during the electrochemical reactions, resulting in Fermi level fluctuations[62,63]. The GC-DFT method adjusts the number of electrons in the systems, thus maintaining the Fermi level for all electrochemical reaction steps. This method essentially provides an accurate description of the electrode-electrolyte interface at a constant electrode potential[64–66].

To account for the combined effect of solvation and applied potentials, we treated the liquid-metal interface as a polarizable continuum using the linearized Poisson-Boltzmann equation, as implemented in VASPsol[67], where the Debye length was set to 3 Å, corresponding to a 1 M concentration of electrolytes with a relative permittivity of 78.4.

In this computational framework, we could tune the potential of the system by changing the number of electrons. The potential versus standard hydrogen electrode ($U_{SHE}$) was calculated as

$$U_{SHE} = (-\mu_e - \Phi_{SHE})/e \qquad (3)$$

where $\mu_e$ and $\Phi_{SHE}$ correspond to the chemical potential of an electron and the work function of the standard hydrogen electrode (SHE), respectively. $\mu_e$ is defined as the Fermi level ($\epsilon_f$) compared to the electrostatic potential of the bulk electrolyte ($V_{bulk}$) (Supplementary Fig. 50), and $\Phi_{SHE}$ was calculated to be 4.43 by RPBE[68]. By changing the number of electrons, one can equate the $\mu_e$ of many states during the

electrochemical reactions, thus maintaining the potential. We set the convergence criteria for $U$ to be $10^{-3}$ V.

The grand canonical electronic energy is calculated as

$$E_{GC-DFT} = E_{DFT} - \Delta n \mu_e - \Delta n V_{bulk} \qquad (4)$$

where $\Delta n$ is the number of electrons adjusted. Thus, $\Delta n$ is positive (negative) when electrons are added (subtracted)[58,69].

### Computational details

Spin polarized DFT calculations were performed using the Vienna Ab initio Simulation Package (version 5.4.4)[70]. The generalized gradient approximation with the revised Perdew-Burke-Ernzerhof (GGA-RPBE) functional[71,72] was used to describe the exchange-correlation interaction, and the D3 method of Grimme with a zero-damping function[73] was applied to include van der Waals interactions. The cutoff energy was set to 400 eV, and the convergence tolerances of energy and force were set to $10^{-4}$ eV and 0.05 eV/Å, respectively. ($2 \times 2 \times 1$) Monkhorst-Pack mesh of $k$-points[74] was sampled. We generated the initial guess of the transition state (TS) of *CO dimerization using the climbing image nudged elastic band (CI-NEB) method[75] with five intermediate images. Subsequently, we employed the dimer method[76] to converge the TS to the saddle point at the specific potential by adjusting the number of electrons.

To simulate the Cu catalyst surfaces, we fully relaxed a bulk structure of face-centered cubic (FCC) Cu. Subsequently, we constructed a three-layered ($4 \times 4$) Cu (100) surface, with the bottom-most layer fixed to the bulk positions. To model the liquid-metal interfaces, we prepared 25 water molecules with one $K^+$ ion[11,77]. We then included a vacuum region of ~12 Å in the z-direction to prevent any spurious interactions between repeating images. To obtain the liquid configuration, we conducted Ab-initio molecular dynamics (AIMD) simulations of a *$CO_2$ adsorbed Cu surface in the NVT ensemble using the Nose-Hoover thermostat for 5 ps, with a time step of 1 fs at 300 K. After confirming that the system had equilibrated, we optimized the last snapshot from the simulation and used it as the starting structural configuration. To model AA/Cu (100), we initially positioned AA approximately 3.5 Å above the surface according to the previous work on AA/Pd (100)[78], and then performed the AIMD simulations. Note that the optimized position of $K^+$ is approximately 6 Å above the surface for both Cu (100) and AA/Cu (100). The AIMD simulations demonstrate that $H_{OX1}$ is most closely positioned to the adsorbed *$CO_2$, suggesting that it is reasonable to assume the proton transfer of $H_{OX1}$ in AA to adsorbates during the $CO_2RR$ (Supplementary Fig. 51).

The grand canonical Gibbs free energy ($G_{GC-DFT}$) was calculated by adding Gibbs free energy correction values ($G_{corr}$) to $E_{GC-DFT}$. The correction values for adsorbates (gaseous molecules) were calculated using harmonic oscillator (ideal gas) approximations at 298.15 K in neutral systems as implemented in the Atomic Simulation Environment (ASE) (Supplementary Table 4)[79]. To correct the gas-phase errors originating from the RPBE functional, we added +0.46 eV to the DFT energy of the $CO_2$ molecule[80].

## Data availability

The data that support the findings of this study are available from the corresponding authors upon request. Source data for the figures in the main text are provided with this paper. Source data are provided with this paper.

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

## Acknowledgements

This work was supported by the National Research Foundation of Korea (NRF) grant funded by the Korean government (MSIT) (NRF-2021R1C1C1013784 and 2021M3D1A2047041, D.-H.N.). S.B. acknowledges the support from the National Research Foundation of Korea (NRF) funded by the Ministry of Education (NRF-2015M3D3A1A01064929, S.B.) and generous supercomputing time from KISTI. S.B. thanks Dr. Changhyeok Choi (University of Toronto) for helpful discussions on grand canonical DFT method. Y.L. acknowledges the support from Basic Science Research Program through the National Research Foundation (NRF) funded by the Ministry of Science and ICT (NRF-2018R1A5A1025594 and 2023R1A2C1003194, Y.L.). Y.L. also acknowledges the support from the BK21 FOUR program through the National Research Foundation (NRF) funded by the Ministry of Education of Korea. Experiments at PLS-II were supported in part by MSIT and POSTECH.

## Author contributions

D.-H.N., Youngu L., and S.B. conceived and supervised the project. J.K. carried out sample synthesis and characterization. M.K. and B.K. assisted sample preparation. T.L. performed electrochemical CO2RR, in situ Raman and operando XAS analysis. J.E. assisted electrochemical CO2RR measurement. W.K., D.B., Yujin L., S.P. contributed to the real time analysis. H.J. and J.P. participated in data collection. S.B. and H.D.J. contributed to the DFT calculation and analysis. All authors discussed the results and contributed to writing the manuscript.

## Competing interests

The authors declare no competing interests.
