## [Peer Review File · Nature Communications]

REVIEWER COMMENTS

Reviewer #1 (Remarks to the Author):

The authors have reported CuNW with ascorbic acid nanoconfined by graphene quantum dots (cAA-CuNW) to achieve the high-rate C₂H₄ with Faradaic efficiency of 60.7% and improved JC₂H₄. Various characterization methods and performed DFT simulations were employed to comprehend the experimental observations. In general, the research topic is intriguing, and the work was conducted competently in many aspects. Nevertheless, certain discussions and conclusions, particularly those pertaining to the simulation part, require careful deduction. I would suggest the authors consider the following comments to further improve the quality of their work before the publication:

1. The interaction between AA and CO₂ in the DFT calculation section lacks clarity. The provided figures, Figure 5b and Figure S22, only illustrate a monodentate structure between HOX1 and CO₂*. Additionally, the coordination of CO₂* with H₂O is also depicted as monodentate. However, it should be noted that AA molecules contain two OH groups apart from HOX1 and HOX2. More calculations are needed to identify the optimal configurations to well describe the studied system, which is crucial for the further understanding of the catalytic system.
2. Explicit solvation model was considered in this work. It might be good to describe the computational method used for this model.
3. In the experimental setup, GQDs are utilized as a mediator to anchor AA onto the Cu surface. However, it is noted that GQDs are absent in the simulation models employed for the DFT calculations. I am curious to know if this absence of GQDs in the simulations would have any impact on the DFT results. If there is no influence, I kindly request an explanation to clarify this matter.
4. In Figure 5a, the transition from *OCCO to 2*CO on AA/Cu(111) appears to be barrierless, indicating that *OCCO is not stable. Based on this observation, it becomes difficult to anticipate an enhancement in *CO dimerization on AA/Cu(111). I suggest the authors to provide the explanation regarding the statement on page 14, line 324-325: "the computational results predicted that AA lowers the activation barrier of *CO dimerization, improving C₂+ selectivity".
5. In Figure 5a, a comparison was made between water deprotonation and AA deprotonation. I kindly request the authors to provide computational details along with the corresponding configurations for this analysis. Additionally, considering the interaction of CO₂ with Cu (111) in the presence of water/[AA+water], it would be valuable for readers to gain insight into the methodology used for incorporating water layers in free adsorption energy calculations.

Reviewer #2 (Remarks to the Author):

This manuscript by Kim et al. reported a study of CO₂ electroreduction on Cu nanowires with ascorbic acid (AA) nanoconfined by graphene quantum dots (cAA-CuNW). Graphene quantum dots were used as a mediator to enable the immobilization and redox reversibility of AA on the Cu electrocatalysts, which showed higher current density and C₂⁺ selectivity for CO₂ reduction than that on pristine Cu nanowires. In situ characterizations using Raman spectroscopy and operando XAS were performed to examine the *CO coverage and Cu chemical state during the reactions. In addition, DFT calculations were used to derive the Gibbs free energy diagram to explain the reaction kinetics, such as the promotion of *CO formation and dimerization by AA/DHA.

Overall, the authors performed a reasonable amount of work to develop a method to immobilize water-soluble AA on electrocatalysts, and then understand the effect on CO₂ electroreduction. Both ex situ and in situ characterizations, as well as computational studies were used to examine the catalyst materials and the possible mechanisms. However, considering the rapid development of the field, the novelty of the method is very limited, the reported CO₂ reduction performance was obtained at a high overpotential, the conclusions are not well supported by the data, and the proposed mechanism is not convincing, as described in detail below. Therefore, I don't think this manuscript can be considered for publication in Nature Communications.

(1) Lack of novelty. The concept of molecular enhancement of CO₂ reduction was summarized and discussed in a paper by the corresponding author (10.1038/s41563-020-0610-2) as well as another review paper in 2020 (10.1038/s41929-020-00512-x). Various immobilized modulators have been demonstrated or proposed, so the example of AA shown in this work cannot provide novel concepts or methodology on this topic.

(2) The main conclusion is unclear. On the one hand, the authors claimed "molecularly enhanced CO₂ mass transport". On the other hand, the authors performed DFT calculations and attributed the improvement to the promotion of *CO formation and dimerization as key reaction steps. What is the role of nanoconfined AA, enhancing CO₂ mass transport or altering the binding of key intermediates? These two possible mechanisms are distinct.

(3) For the comparison of the electrodes in Supplementary Fig. 15, the current density on the CuNW and AA-CuNW electrodes reached a plateau at more negative potentials, which should be caused by CO₂ mass transport limitation. In contrast, the current density on the G-CuNW and cAA-CuNW electrodes both showed a different potential dependence and continued to increase rapidly with the overpotential, indicating a mitigation of the CO₂ mass transport limitation with the presence of graphene materials. While AA has some promotional effect, the added graphene quantum dots may have a similarly

important effect. The author must carefully consider this and quantify their wetting properties and resulting CO₂ mass transport on these electrodes.

(4) The electrodes for comparison also had different surface roughness (shown by TEM images in Figure 1). Does the roughness impact the CO₂ reduction current density? The authors must measure the electrochemically active surface area (ECSA) of the electrodes and compare ECSA-normalized activity to exclude the possible effect of surface roughness.

(5) The authors pointed out that “AA can react with CO₂ and be oxidized to dehydroascorbic acid (DHA) with proton and electron donation”. It is unclear that whether AA directly participates in the CO₂ reduction reaction, or just serves as an electron or proton shuttle?

(6) What is the mechanism of the nanoconfinement by graphene quantum dots, just physically covering the surface so that the AA cannot diffuse away? What adsorption or interaction do the AA molecules have with the graphene or the Cu nanowire surface, physisorption, chemisorption, or others?

(7) Stability of the nanoconfined AA during long-term electrolysis. As the authors indicated, “AA is easily dissolved into the electrolyte due to its high solubility in aqueous solutions”. Figure 2g showed that there was still DHA extracted in the solution from cAA-CuNW even with Nafion ionomer coating after 60 min, so the AA/DHA was detaching slowly from the surface. Thus, how long can the effect of nanoconfined AA last? Long-term electrolysis test with the quantification of AA/DHA in the electrode after the test must be shown.

(8) The DFT calculations did not consider the cations in the electrolyte. However, recent study demonstrated that CO₂ electroreduction will not occur on Cu surface without metal cations (10.1038/s41929-021-00655-5). The authors showed CO₂ reduction on Cu(111) surface without the presence of any cations. Does this result conflict with the literature (10.1038/s41929-021-00655-5)?

(9) The authors reported a maximum J_{C₂H₄} of 539 mA/cm² at -1.55 V vs. RHE, which is 2.9-fold higher than the highest J_{C₂H₄} of pristine CuNW with 184 mA/cm² at -1.39 V vs RHE. However, the two current densities were measured at different potentials, so the comparison is unfair. Similarly, Supplementary Table 1 shows a comparison of various Cu-based CO₂RR catalysts for high-current ethylene production. What is the overpotential for each case? Without the inclusion of applied potentials in the table, such comparison is unfair and even misleading.

Reviewer #3 (Remarks to the Author):

In this manuscript, Kim et al reported that ascorbic acid (AA) can promote the C-C coupling by using AA-graphene quantum dots decorated Cu nanowires (cAA-CuNW), which facilitates the CO₂ electroreduction to ethylene production with higher Faradaic efficiency and current density. By using in situ characterization, the authors try to understand the role of AA on Cu for the boosted mechanism. Overall, the idea that using organic molecules to activate/boost adsorption of CO₂ is not novel, this work provides some fresh and interesting findings and conclusions for the related fields. However, some fundamental grounds have not been well illustrated. Current manuscript is promising but very preliminary. Therefore, it should be better for this manuscript to do major revision, and then this work will be more influential.

Major comments:

1. Although the authors tried to explain the role of AA and GQDs for the CO₂ electroreduction, I still feel confused how these two species work in the system. The authors should clearly explain the behaviour of corresponding components. For example, AA could be oxidized even without purging of CO₂. Therefore, will the AA oxidized form (i.e., DHA) has boosting role for CO₂ reduction?
2. The authors should provide more supportive and direct evidence to illustrate the capture of CO₂ with the help of AA. Some works from Lin Zhuang group (ACS Catalysis, 2022, 12, 1004-1011; ACS Energy Letter, 2022, 7, 4045-4051.) tried to utilized in situ FTIR to monitor the adsorption and activation of CO₂ molecules with the help of some organic molecules.
3. In the materials characterization part, what kind of TEM grid the authors were using? In fact, EDS is not accurate to explain the ratio of C/N/O/Cu, as most TEM grid can also false signals on EDS with the aforementioned elements. Similar situations also happen on XPS measurement. Therefore, the authors should carefully treat these data and results, and discuss them critically. Otherwise, it is suggested to use more accurate analytical methods to draw out the conclusion about non-metal distribution around Cu. Also, according to the TEM images (Fig S4), the outer-shell (3~5nm) is almost CuO, which is contradictory to the discussion that O barely detected.
4. For the catalysis part, the authors should provide more post-catalysis characterizations including TEM and XRD to support the discussion the authors mentioned. It is also recommended that the authors check all catalytic performance data, as some error bars are >10% (relative error). Therefore, these data may not be very convincing to present the trend and differences between all samples.
5. It is good that the authors utilized in situ Raman to quantitatively compare the ratio of C=Obridge and C=Otop peaks and correlate it to the CO coverage. However, according to Fig. 3, the best ethylene production potential lies below -0.8V vs RHE. Therefore, why did the author stop the in situ Raman testing at -0.8V vs RHE? It is suggested to collect all spectra alongside the shifting of the biased potential, and then find the trend.
6. For the DFT calculations, please explain why Cu(111) was utilized for modelling? According to the Fig. 1, TEM images exhibited the Cu (100). Multifacets of Cu could also be found in Fig. S4-S5.

7. The authors should provide the whole pathway gibbs free energy diagram for CO₂ reduction towards CO and ethylene respectively, rather than providing some data on CO adsorption and OCCO. Also, HER should also be discussed. For the last paragraph of result and discussion (Considering that the activation energy is xxx), there is no systematic data in this manuscript to support that conclusion.

Minor comments:

1. Some symbols need to be checked. Please keep the unified expressions for the whole manuscript.
2. The introduction and organization of the whole manuscript should be rearranged.

Author Actions in light of Reviewers Comments

Title: Vitamin C-induced CO₂ capture enables high-rate ethylene production in CO₂ electroreduction

Authors: Jongyoun Kim[†], Taemin Lee[†], Hyun Dong Jung[†], Minkyoung Kim, Jungsu Eo, Byeongjae Kang, Hyeonwoo Jung, Jaehyoung Park, Daewon Bae, Yujin Lee, Sojung Park, Wooyul Kim, Seoin Back*, Youngu Lee* and Dae-Hyun Nam*

Reviewer #1

The authors have reported CuNW with ascorbic acid nanoconfined by graphene quantum dots (cAA-CuNW) to achieve the high-rate C₂H₄ with Faradaic efficiency of 60.7% and improved J_{C₂H₄}. Various characterization methods and performed DFT simulations were employed to comprehend the experimental observations. In general, the research topic is intriguing, and the work was conducted competently in many aspects. Nevertheless, certain discussions and conclusions, particularly those pertaining to the simulation part, require careful deduction. I would suggest the authors consider the following comments to further improve the quality of their work before the publication:

➤ Response:

We appreciate your insightful comments about our work. We demonstrate that the redox activity of molecular additives promotes CO₂-to-*CO conversion and *CO dimerization, which lead to high C₂H₄ productivity. In addition, our DFT simulations reveal that ascorbic acid (AA) on Cu can promote *CO formation and dimerization through favorable electron

and proton transfer and strong hydrogen bonding. In response to your comments, we have improved the quality of our work by revising DFT simulations and providing more detailed information about our computational methods. Detailed point-by-point responses are as follows.

1. The interaction between AA and CO₂ in the DFT calculation section lacks clarity. The provided figures, Figure 5b and Figure S22, only illustrate a monodentate structure between H_{OX1} and CO₂. Additionally, the coordination of CO₂* with H₂O is also depicted as monodentate. However, it should be noted that AA molecules contain two OH groups apart from H_{OX1} and H_{OX2}. More calculations are needed to identify the optimal configurations to well describe the studied system, which is crucial for the further understanding of the catalytic system.*

➤ **Response:**

We appreciate the reviewer's comment. In the revised computational results, we obtained the liquid configuration from Ab-initio molecular dynamics (AIMD) simulations instead of using the static hexagonal water layer as in the original manuscript. To address the reviewer's comment, we plotted the bond length between the hydrogen atoms in four distinct OH groups (H_{OX1~4}) in AA, and the oxygen atom near the electrolyte (O_e) of *CO₂ from the last 3 ps of AIMD simulations (**Fig. R1**). We found that the average bond length of H_{OX1}-O_e was approximately 1.74 Å indicating a favorable formation of a hydrogen bond, while the distances to other H atoms are much farther. For the reviewer's reference, we added the atomic configurations of Cu (100) and AA/Cu (100) in **Fig. R2**.

➤ **Modification:**

(Manuscript page 25 line 563-565) We update the calculation details demonstrating the bond length profile.

(Supplementary information page 46) Fig. R2 has been added as Supplementary Fig. 43.

(Supplementary information page 50) Fig. R1 has been added as Supplementary Fig. 47.

Fig. R1 | The bond lengths between H_{OX1-4} in AA and O_e of *CO₂ measured during the last 3 ps of AIMD simulations.

Fig. R2 | Atomic structures of *CO₂ adsorption on **a**, AA/Cu (100) and **b**, Cu (100). Color code: black (C), white (H), yellow (H), red (O), purple (K), and orange (Cu). Yellow H atoms are considered for the protonation during the CO₂RR.

2. *Explicit solvation model was considered in this work. It might be good to describe the computational method used for this model.*

➤ **Response:**

Following the reviewer's comment, we provide further details regarding the explicit solvation model. In our previous simulation in the original manuscript, we used a static hexagonal water layer with one hydronium ion, which has been widely applied in numerous computational studies on electrochemical reactions on FCC (111) facet (*Nature* **577**, 509-513 (2020), *Nat. Energy* **5**, 478-486 (2020), *J. Phys. Chem. Lett.* **12**, 5193–5200 (2021)).¹⁻³

In the updated calculations, we performed AIMD simulations using 25 water molecules and one K⁺ ion (*Nat. Catal.* **4**, 654-662 (2021), *Nat. Commun.* **13**, 5482 (2022))^{4,5} to construct the solid-liquid interface (*ACS Catal.* **12**, 11530-11540 (2022), *J. Phys. Chem. C* **126**, 7841-7848 (2022)).^{6,7} After confirming the system had equilibrated, we used the last snapshot of the AIMD simulations as the starting structural configuration. Note that the position of the metal cation is crucial, as its interaction with adsorbates depends on its displacement (*J. Am. Chem. Soc.* **145**, 1897-1905 (2023), *J. Am. Chem. Soc.* **145**, 19601-19610 (2023)),^{8,9} and we confirmed that the optimized position of K⁺ is approximately 6 Å above the surface for both Cu (100) and AA/Cu (100).

➤ **Modification:**

(Manuscript page 24-25 line 551-563) We include the aforementioned details regarding the explicit solvation model in the Methods section.

(Manuscript page 33 line 754-761) We add references for the explicit solvation model.

3. In the experimental setup, GQDs are utilized as a mediator to anchor AA onto the Cu surface. However, it is noted that GQDs are absent in the simulation models employed for the DFT calculations. I am curious to know if this absence of GQDs in the simulations would have any impact on the DFT results. If there is no influence, I kindly request an explanation to clarify this matter.

➤ **Response:**

It was reported that functional groups in graphene quantum dots (GQDs) can affect CO₂ reduction reaction (CO₂RR) (Fig. R4).¹⁰ Zhang et al. controlled the functional groups of GQDs by the oxidation or reduction of pristine GQDs. Hydroxyl group-dominant GQDs (reduced GQDs and reduced-oxidized GQDs) exhibited high selectivity and partial current density of CH₄ compared to those of carboxyl group-dominant GQDs (pristine GQDs and oxidized GQDs) (Fig. R4). This indicates that electron donating groups in GQDs promote CO₂RR by maintaining higher charge density, while electron withdrawing groups have no positive effects on CO₂RR.

However, in our work, the effect of GQDs on CO₂RR is not dominant for CuNW with AA nanoconfined by GQDs (cAA-CuNW) because most GQDs lost their functional groups by the reduction during AA introduction and the types of GQD functional groups are less controlled. To confirm the CO₂RR activity of GQDs in cAA-CuNW, we compared CO₂RR of pristine GQDs and GQDs with nanoconfined AA (AA+GQDs) (Supplementary Fig. 25). Pristine GQDs showed 68% H₂ FE, 22% CH₄ FE, and 9% CO FE at -3 V (vs RHE, non-iR corrected). However, GQDs with nanoconfined AA exhibited dominant hydrogen evolution reaction (HER) with 91% H₂ FE at -3 V (vs RHE, non-iR corrected) due to the low amount of oxygenated functional groups. We think that GQDs have less effect on CO₂RR, and act as a mediator to anchor AA onto Cu surface. Therefore, we excluded GQDs from the simulation model because the presence of GQDs is not expected to have a significant impact on the DFT results.

➤ **Modification:**

(Supplementary information page 3) We add explanation of the impact of the absence of GQDs in the DFT results.

T. Zhang et al., *Nat. Commun.* **12**, 5265 (2021)

Fig. R4 | Effect of GQDs on CO₂RR. a, O 1s and **b**, C 1s XPS data for pristine GQD (p-GQD), oxidized-GQD (o-GQD), reduced-GQD (r-GQD), and reduced o-GQD (ro-GQD), respectively. **c**, Faradaic efficiency (FE) and partial current density of CH₄ for different GQDs. Reproduced from T. Zhang et al., *Nat. Commun.* **12**, 5265 (2021).¹⁰

4. In Figure 5a, the transition from *OCCO to 2*CO on AA/Cu(111) appears to be barrierless, indicating that *OCCO is not stable. Based on this observation, it becomes difficult to anticipate an enhancement in *CO dimerization on AA/Cu(111). I suggest the authors to provide the explanation regarding the statement on page 14, line 324-325: “the computational results predicted that AA lowers the activation barrier of *CO dimerization, improving C₂₊ selectivity.”.

➤ **Response:**

In our updated calculations, we observed the activation barrier for *CO dimerization on both Cu (100) and AA/Cu (100) as shown below. All relevant text and figures have been updated accordingly.

➤ **Modification:**

(Manuscript page 17 line 395-403) We revise explanation about the activation barrier for *CO dimerization on both Cu (100) and AA/Cu (100) based on our reevaluated calculations.

(Manuscript page 31 line 705-713) We add references for explanation about the activation barrier for *CO dimerization.

(Manuscript page 40) We update the activation barrier plot for *CO dimerization (Fig. R5) in Fig. 5a.

Fig. R5 | The Gibbs free energy diagram of *CO dimerization at 0 V (vs RHE).

5. In Figure 5a, a comparison was made between water deprotonation and AA deprotonation. I kindly request the authors to provide computational details along with the corresponding configurations for this analysis. Additionally, considering the interaction of CO₂ with Cu (111) in the presence of water/[AA+water], it would be valuable for readers to gain insight into the methodology used for incorporating water layers in free adsorption energy calculations.

➤ **Response:**

Following the reviewer's comment, we provide further details regarding the atomic structures we used for the analysis. To calculate the Gibbs free energy of protonation process, we calculated the energy difference of the initial and final state.

Fig. R6 illustrates *CO₂ protonation to *COOH using two different sources (H₂O or AA), where we have one initial state and two final states. In the Supplementary Information, we added the reaction expressions for all step reactions considered in this work.

➤ **Modification:**

(Manuscript page 24-25 line 540-571) We revise the explanation on our computational method for our reevaluated DFT calculations.

(Supplementary information page 2-3) We add the (R1a, R2a, R3a, R4a, R5a, R6a) and (R1b, R2b, R2c, R3b, R4b, R5b, R5c, R6b, R6c) to represent the modeled reactions for Cu (100) and AA/Cu (100), respectively.

Fig. R6 | The atomic structures of the initial and final states of $^*\text{CO}_2$ protonation to $^*\text{COOH}$ formation on AA/Cu (100).

Reviewer #2

*This manuscript by Kim et al. reported a study of CO₂ electroreduction on Cu nanowires with ascorbic acid (AA) nanoconfined by graphene quantum dots (cAA-CuNW). Graphene quantum dots were used a mediator to enable the immobilization and redox reversibility of AA on the Cu electrocatalysts, which showed higher current density and C₂₊ selectivity for CO₂ reduction than that on pristine Cu nanowires. In situ characterizations using Raman spectroscopy and operando XAS were performed to examine the *CO coverage and Cu chemical state during the reactions. In addition, DFT calculations were used to derive the Gibbs free energy diagram to explain the reaction kinetics, such as the promotion of *CO formation and dimerization by AA/DHA.*

Overall, the authors performed a reasonable amount of work to develop a method to immobilize water-soluble AA on electrocatalysts, and then understand the effect on CO₂ electroreduction. Both ex situ and in situ characterizations, as well as computational studies were used to examine the catalyst materials and the possible mechanisms. However, considering the rapid development of the field, the novelty of the method is very limited, the reported CO₂ reduction performance was obtained at a high overpotential, the conclusions are not well supported by the data, and the proposed mechanism is not convincing, as described in detail below. Therefore, I don't think this manuscript can be considered for publication in Nature Communications.

➤ **Response:**

We appreciate your constructive feedback on our work. Based on your insightful comments, we have taken extensive revisions to address each concern with point-by-point response. We provide the role of AA more clearly and confirm that the redox activity of AA promotes CO₂-to-*CO conversion and *CO dimerization, contributing to high C₂H₄ productivity. Furthermore, we discovered that AA contributes to efficient CO₂RR of cAA-CuNW in low CO₂ concentration, which can be extended to CO₂RR of flue gas (**Fig. R8**). We also demonstrate the exploitation of the nanoconfinement effect *via* GQDs to ensure reversible redox cycling of AA in aqueous electrolyte (**Fig. R10**) and verified the stable C₂H₄ production of cAA-CuNW (**Fig. R20**). Detailed point-by-point responses are as follows.

1. *Lack of novelty. The concept of molecular enhancement of CO₂ reduction was summarized and discussed in a paper by the corresponding author (10.1038/s41563-020-0610-2) as well as another review paper in 2020 (10.1038/s41929-020-00512-x). Various immobilized modulators have been demonstrated or proposed, so the example of AA shown in this work cannot provide novel concepts or methodology on this topic.*

➤ **Response:**

This study newly reports that redox activity of AA can achieve high C₂H₄ productivity in CO₂RR by promoting (1) CO₂-to-*CO conversion and (2) *CO dimerization, which have been verified by CO₂RR, real-time analysis, and theoretical computation (**Fig. R7**). To facilitate the continuous redox cycle for CO₂ capture with proton and electron transfer, we developed nanoconfinement to immobilize this redox-active molecules (AA) using GQDs. We achieved high-rate C₂H₄ production with high partial current density of C₂H₄ (J_{C₂H₄}) in CO₂RR. Furthermore, we again prove the novelty of our approach by demonstrating the efficient CO₂RR of cAA-CuNW even in low CO₂ concentration (**Fig. R8**).

In previous CO₂RR studies using molecular approaches, the role of immobilized modulators can be categorized into (i) enhancing CO₂ mass transport for *CO formation (e.g. Nafion,¹¹ phenylpyridinium,¹² polyaniline (PANI)¹³), (ii) optimizing hydrophobicity for *CO formation (e.g. polyvinylidene fluoride (PVDF),¹⁴ alkanethiol^{15,16}), (iii) promoting *CO dimerization (e.g. pyridinium additives,^{1,17} poly(acrylamide)¹⁸), and (iv) *CHO stabilization (e.g. amino acids¹⁹). They contribute to increasing *CO coverage by enhancing CO₂ mass transport (increased local CO₂ concentration) or optimizing hydrophobicity. In this work, nanoconfined AA electrochemically promotes (1) CO₂-to-*CO conversion by facilitating electron and proton transfer from AA/DHA redox cycle and (2) *CO dimerization. This is different with previous approaches. Novelty of our work is explained more in detail as follows:

(1) CO₂-to-*CO conversion: With comprehensive studies including AA/DHA redox cycle, CO₂RR performances, *in situ* Raman analysis, and DFT calculation, it was confirmed that *CO formation is promoted in the presence of AA. We further proved the effect of nanoconfined AA on CO₂-to-*CO conversion by comparing the CO₂RR of pristine CuNW (p-CuNW) and cAA-CuNW at low CO₂ concentration (**Fig. R8**). In low CO₂ concentration, HER is dominant because of limited CO₂ mass transport. p-CuNW

showed a dramatic increase of H₂ FE as the CO₂ ratio decreased in CO₂+Ar mixed gas (H₂ FE of 53.3% and C₂H₄ FE of 15.6% at the CO₂ ratio of 33%). In contrast, cAA-CuNW exhibited H₂ FE of 19.6% and C₂H₄ FE of 41.8% even at the CO₂ ratio of 33%. Although local CO₂ concentration decreased, CO₂RR of cAA-CuNW was efficient. Unlike the previous approaches for CO₂RR at low CO₂ concentration (such as increasing local CO₂ concentration^{20,21} or preventing HER²²), nanoconfined AA lowers the activation energy of CO₂-to-*CO conversion by efficient proton and electron transfer from AA/DHA redox. Therefore, C₂H₄ selectivity of cAA-CuNW can be maintained by efficient CO₂-to-*CO conversion even at low CO₂ concentration and this is different with other molecular approaches. The merit of our method is that nanoconfined AA (for promoting CO₂-to-*CO conversion) can be combined with other molecules (for enhancing CO₂ mass transport) to boost the production rate of CO₂RR.

(2) *CO dimerization: We also confirmed that nanoconfined AA promotes *CO dimerization. Regarding C₂H₄ (*CO dimerization)/CH₄ (*CO hydrogenation) formation mechanism,²³ comparing the ratio of C₂H₄ FE to CH₄ FE (C₂H₄ FE/CH₄ FE) enables to confirm whether the *CO on catalyst surface is favorable to dimerization or not. In **Fig. R9**, C₂H₄ FE/CH₄ FE was much higher in cAA-CuNW. This indicates that *CO prefers dimerization in cAA-CuNW. *CO binding mode analysis by *in situ* Raman spectroscopy revealed that cAA-CuNW has optimal ratio of bridge bound *CO (CO_{bridge}) and atop bound *CO (CO_{atop}) for C-C coupling toward C₂ products. Furthermore, DFT calculation of the activation energy clearly shows the promotion of *CO dimerization in the presence of AA (Fig. 5a).

Our work also provides a guideline for harnessing small molecules in CO₂RR. Dissolution is critical issue when we introduce molecular additives to the heterogeneous catalyst. Therefore, most studies utilize large molecules, not easily soluble in aqueous electrolytes. Immobilizing the small molecules was enabled in the limited cases such as molecules which contain anchoring groups (amines or thiols). However, we achieved immobilization and redox stabilization of small redox-active molecules through the nanoconfinement of GQDs.

➤ **Modification:**

(Manuscript page 2 line 24-25 and 31-33) We revise the abstract of the manuscript to highlight the role of nanoconfined AA.

(Manuscript page 3-4 line 55-72 and 93-94) We revise the introduction of the manuscript to address the novelty of nanoconfined AA by providing background of other molecular approaches and explaining the role of nanoconfined AA more clearly.

(Manuscript page 12 line 262-264) We add explanation about the role of nanoconfined AA on *CO dimerization.

(Manuscript page 13 line 286-293) We add explanation about the CO₂RR of p-CuNW and cAA-CuNW by controlling the CO₂ ratio of CO₂+Ar mixed gas.

(Manuscript page 18-19 line 425-428 and 433-435) We revise the conclusion the manuscript to highlight the role of nanoconfined AA.

(Manuscript page 22-23 line 509-511) We provide experimental details of CO₂RR in CO₂+Ar mixed gas.

(Manuscript page 26-28 line 596-597 and 615-627) We add references for explanation about previous CO₂RR studies using molecular approaches in the introduction section.

(Manuscript page 38) We add the results of CO₂RR under CO₂+Ar mixed gas (Fig. R8a and R8b) in Fig. 3g and 3h. Original graphs has been moved to Supplementary Fig. 29.

(Supplementary information page 31) Fig. R9 has been added as Supplementary Fig. 28.

(Supplementary information page 37) Fig. R8c–d have been added as Supplementary Fig. 34.

Role of nanoconfined AA for high C₂H₄ productivity in CO₂RR

Fig. R7 | An overview of the role of nanoconfined AA for CO₂RR with high C₂H₄ productivity.

Table R1 | Summary of the role and CO₂RR performance of the molecular additives for Cu catalysts.

Molecular additives			Verification		Integration method	Material	CH ₄ FE (%)	C ₂ H ₄ FE (%)	J _{C₂H₄} (mA/cm ²)	Ref.
Role	Type	Effect	Experiment	Theory						
(1) Promote CO ₂ -to-*CO conversion (2) Promote *CO dimerization	Nanoconfined ascorbic acid	△, □, ☆	○	○	Nanoconfinement by GQDs	p-CuNW (100% CO ₂)	54.4	184	This work	
						cAA-CuNW (100% CO ₂)	60.7	539		
						p-CuNW (33% CO ₂)	15.6	49.8		
						cAA-CuNW (33% CO ₂)	41.8	129.6		
Enhance CO ₂ mass transport for *CO formation	Nafion ionomer	△, □	○	○	Solution mixing	Cu CIPH	53	380	11	
						CIBH	60	930		
	Phenylpyridinium-based copolymer	△	×	○	Ring-opening metathesis polymerization	Cu foil	22.3	1.16	12	
						Cu foil + Phenylpyridinium-based copolymer	55.6	2.59		
	Polyaniline (PANI)	△	○	×	Drop casting	Cu	20	7.2	13	
						Cu + PANI	48.8	16.93		
Optimize hydrophobicity for *CO formation	Polyvinylidene fluoride (PVDF)	△	○	×	Drop casting	CuO NPs	16.7	1.4	14	
						CuO NPs + PVDF	40.6	3.8		
	Alkanethiol 1	△	○	×	Thiol anchoring	Dendritic Cu	9	2.7	15	
						Dendritic Cu + alkanethiol	56	16.8		
	Alkanethiol 1	△	○	×	Thiol anchoring	Cu	26.3	52.6	16	
						Cu + 1-octadecanethiol	35.6	44.5		
Promote *CO dimerization	N-substituted pyridinium additives	△	×	×	Electrochemical deposition	Cu foil	12.4	0.55	17	
						Cu foil + N-tolylpyridinium chloride	40.5	0.41		

	N-substituted pyridinium additives	△	○	○	Electrochemical deposition	Cu	43.9	160	1	
						Cu + pyridinium oligomer	71.5	230		
	Poly(acrylamide)	△	x	○	Electrodeposition	Cu foam	13	7.2	18	
						Cu foam + P-Acrylamide	26	15.6		
*CHO stabilization	Amino Acids	△	x	○	Amine anchoring	Cu foil	16.1	9.5	0.37	19
						Cu foil + glycine	32.1	24.0	0.89	

△: increase of C₂H₄ FE, □: increase of J_{C₂H₄} over 300 mA/cm², ☆: efficient CO₂RR in CO₂ deficient environment

Fig. R8 | Investigation of CO₂RR at low CO₂ concentration. Gaseous product FEs and total current densities of **a**, p-CuNW and **b**, cAA-CuNW by the CO₂RR according to the CO₂ ratios in CO₂+Ar mixed gas. Total product FEs and total current densities of **c**, p-CuNW and **d**, cAA-CuNW by the CO₂RR according to the CO₂ ratios in CO₂+Ar mixed gas. All measurements were conducted under the applied potential of -2.2 V (vs RHE, non-*i*R corrected) with 1 M KOH electrolyte. Gas flow rate was fixed to 60 sccm.

Fig. R9 | Comparison of the C₂H₄ and CH₄ selectivity of CuNWs. C₂H₄ FE/CH₄ FE in p-CuNW, G-CuNW, AA-CuNW, and cAA-CuNW according to applied potentials.

2. The main conclusion is unclear. On the one hand, the authors claimed “molecularly enhanced CO₂ mass transport”. On the other hand, the authors performed DFT calculations and attributed the improvement to the promotion of *CO formation and dimerization as key reaction steps. What is the role of nanoconfined AA, enhancing CO₂ mass transport or altering the binding of key intermediates? These two possible mechanisms are distinct.

➤ **Response:**

In **Table R1**, we categorize the previous molecular approaches for *CO formation as enhancing CO₂ mass transport (local CO₂ concentration) and optimizing the hydrophobicity. In this regard, the expression “molecularly enhanced CO₂ mass transport” is not the direction pursued in our work, because nanoconfined AA increase *CO coverage by promoting CO₂-to-*CO conversion, different with the approaches for enhancing CO₂ mass transport in previous works.

For better clarification, we revised our previous expression of ‘molecularly enhanced CO₂ mass transport’ to ‘molecularly enhanced CO₂-to-*CO conversion’. In this context, the salient properties of nanoconfined AA are (1) promoting *CO formation onto the catalyst by facilitating electron and proton transfer from AA/DHA redox cycle, and (2) promoting *CO dimerization. In fact, an increase in *CO coverage and optimized *CO binding modes in the presence of nanoconfined AA were confirmed through *in situ* Raman analysis (Fig. 4a–h).

To investigate whether the promotion of *CO formation is originated from the electrochemical redox of nanoconfined AA, we compared the cyclic voltammetry (CV) of AA and nanoconfined AA on GQDs under N₂ and CO₂ supply (**Fig. R10**). When AA was dissolved in the electrolyte, the oxidation of AA was observed at 0.23 and 0.26 V (vs Ag/AgCl) in both N₂ and CO₂ supply. However, the paired reduction peak was not observed in the reverse scans, indicating that AA was irreversibly converted to DHA (**Fig. R10a**). We expect this to limit the promotion of CO₂-to-CO* conversion; the enol group in AA was oxidized to carbonyl group, which does not donate electrons and protons to CO₂. In contrast, nanoconfined AA on GQDs shows the paired reduction peak (–0.36 V vs Ag/AgCl) in both N₂ and CO₂ supply (**Fig. R10b-d**). This indicates that reversible AA/DHA redox is achieved in nanoconfined AA on GQDs by enhancing electron and proton transfer to promote *CO formation on the catalyst. When we see the increased *CO

coverage in the *in situ* Raman spectroscopy of cAA-CuNW, we think that nanoconfined AA can promote CO₂-to-*CO conversion.²⁴⁻²⁶ Based on these results, we present a schematic diagram of the electrochemical behavior of nanoconfined AA and CO₂RR in **Fig. R11**.

Nanoconfined AA can also control the bindings of *CO intermediates. The high C₂H₄ FE/CH₄ FE of cAA-CuNW indicates that cAA-CuNW induces *CO dimerization favorable intermediates bindings (**Fig. R9**). We performed further analysis of the *CO binding mode of G-CuNW and AA-CuNW in *in situ* Raman spectroscopy and compared the *CO bindings of all CuNWs during CO₂RR (**Fig. R12** and **R13**). cAA-CuNW exhibited optimized the CO_{bridge}/CO_{atop} ratio even when the potential increases up to -0.8 V (vs RHE, non-iR corrected), while p-CuNW and G-CuNW show an excessive CO_{atop} or CO_{bridge} as the potential increases. As a result, nanoconfined AA in cAA-CuNW not only induces high *CO coverage on the catalyst surface, but also plays a role in controlling *CO binding that promote C₂₊ chemical production.

Therefore, we suggest the role of nanoconfined AA for C₂H₄ production of CO₂RR as follows; i) promotion of CO₂-to-*CO conversion by ensuring electron and proton transfer from redox cycle, ii) promotion of *CO dimerization by altering the bindings of *CO intermediate.

➤ **Modification:**

(Manuscript page 3, 4, 5, 11, 12, 15, 18, 36, 37, and 38 line 49-50, 53, 86-87, 100, 104, 261, 284-285, 353, 416, 420-421, 799-800, 811, and 816-817) We modify the content from “CO₂ mass transport” to “CO₂-to-*CO conversion” to clearly suggest the role of nanoconfined AA for C₂H₄ production.

(Manuscript page 9 line 200-206) We add the explanation about the electrochemical analysis of AA and nanoconfined AA on GQDs in N₂ and CO₂ supply.

(Manuscript page 14-15 line 317-339) We revise the explanation about the *in situ* Raman spectroscopy of CuNWs and the analysis of CO_{bridge}/CO_{atop} ratio.

(Manuscript page 39) We provide the *in situ* Raman spectroscopy of G-CuNW and AA-

CuNW during CO₂RR (Fig. R12) in Fig 4c-f.

(Manuscript page 39) We provide the comparison of integral area ratios of CO_{bridge} and CO_{atop} (Fig. R13) in Fig. 4i.

(Supplementary information page 19-20) Fig. R10-R11 has been added as Supplementary Fig. 16-17.

Fig. R10 | Redox behavior of AA and nanoconfined AA on GQDs. a, CV plot of GCE with AA dissolved in 0.1 M KHCO₃ electrolyte. **b,** CV plot of GCE coated with nanoconfined AA on GQDs in 0.1 M KHCO₃ electrolyte. 4th cycle of the CV plot of GCE coated with nanoconfined AA on GQDs with **c,** N₂ and **d,** CO₂ supply.

Fig. R11 | Schematic illustration of our strategy for CO₂ capture in atmosphere. The redox behavior of nanoconfined AA on GQDs contributes to CO₂ capture by promoting CO₂-to-*CO conversion during CO₂RR.

Fig. R12 | Real-time observation of *CO bindings from CuNWs. *In situ* Raman spectra obtained during the CO₂RR under different applied potentials from **a–b**, p-CuNW, **c–d**, G-CuNW, **e–f**, AA-CuNW, **g–h**, cAA-CuNW region of 200-700 cm⁻¹ (top) and 1,800-2,400 cm⁻¹ (bottom).

Fig. R13 | Comparison of *CO binding modes of CuNWs. Comparison of integral area ratios of CO_{bridge} and CO_{atop} for CuNWs in *in situ* Raman spectra.

3. For the comparison of the electrodes in Supplementary Fig. 15, the current density on the CuNW and AA-CuNW electrodes reached a plateau at more negative potentials, which should be caused by CO₂ mass transport limitation. In contrast, the current density on the G-CuNW and cAA-CuNW electrodes both showed a different potential dependence and continued to increase rapidly with the overpotential, indicating a mitigation of the CO₂ mass transport limitation with the presence of graphene materials. While AA has some promotional effect, the added graphene quantum dots may have a similarly important effect. The author must carefully consider this and quantify their wetting properties and resulting CO₂ mass transport on these electrodes.

➤ **Response:**

We further investigated CO₂RR of G-CuNW up to -1.89 V (vs RHE) and confirmed that the current density on G-CuNW reached a plateau and HER significantly increased at -1.74 V (vs RHE) (**Fig. R14**). The total current density of G-CuNW saturated up to 465 mA/cm², higher than that of CuNW (348 mA/cm²) and AA-CuNW (431 mA/cm²) but is still much lower than that of cAA-CuNW (957 mA/cm²). Enhanced total current density of G-CuNW can be attributed to the catalytic activity of GQDs.

GQDs can enhance the electrochemical kinetics for high capacitance,²⁷ and have been utilized as electrocatalysts in various catalytic reactions.^{10,28,29} Especially in CO₂RR, the catalytic activity of GQDs is determined by the types of the functional group; the electron donating groups promote CO₂RR by maintaining higher charge density, while the electron withdrawing groups exhibit no positive effects on CO₂RR.¹⁰ To confirm the effect of GQDs on CO₂RR activity, we compared CO₂RR of pristine GQDs and GQDs with nanoconfined AA on GQDs (Supplementary Fig. 25). Pristine GQDs showed 68% H₂ FE, 22% CH₄ FE, and 9% CO FE at -3 V (vs RHE, non-iR corrected), implying that GQDs can function as CO₂RR active sites because of oxygenated functional groups (Fig. 2b). However, since the types of functional groups (electron donating and electron withdrawing groups) were not perfectly controlled for the GQDs, the maximum total current density was not higher than cAA-CuNW.

Furthermore, we investigated the impact of GQD properties on CO₂ mass transport. Functional groups in graphene-based materials hinder the formation of gas diffusion channels.^{30,31} Furthermore, GQD is hydrophilic due to large amount of oxygen containing

functional groups on edge site.³² To determine the effect of the hydrophilic GQDs on the wetting properties of CuNWs, we measured the water contact angle of each CuNW (**Fig. R15**). The water contact angles of G-CuNW and cAA-CuNW were slightly lower than others due to hydrophilic properties of GQDs. Considering that hydrophobicity is important for CO₂ mass transport,^{14,16} the hydrophilic properties of GQDs are not beneficial for high current density CO₂RR. Therefore, we think that the enhanced CO₂RR productivity of cAA-CuNW are attributed to the promotion of CO₂-to-*CO conversion by nanoconfined AA rather than the effects caused by GQDs.

➤ **Modification:**

(Manuscript page 10-11 line 233-235 and 239-241) We add explanation about the effect of GQDs on CO₂RR.

(Manuscript page 30 line 675-676) We add references for explanation about the hindered gas diffusion of GQDs.

(Supplementary information page 27) Fig. R14 has been added as Supplementary Fig. 24.

(Supplementary information page 29) Fig. R15 has been added as Supplementary Fig. 26. Detail explanation about the contribution of GQDs to CO₂ mass transport was described.

(Supplementary information page 56) We add references for the explanation about the impact of hydrophilic properties of GQDs on high current density CO₂RR.

Fig. R14 | Investigation of CO₂RR of G-CuNW at extended potential range. Gaseous product FEs and total current densities for G-CuNW up to -1.89 V (vs RHE).

Fig. R15 | Wetting properties of the CuNWs. Water contact angle of **a**, CuNW, **b**, G-CuNW, **c**, AA-CuNW, and **d**, cAA-CuNW on PTFE substrates.

4. The electrodes for comparison also had different surface roughness (shown by TEM images in Figure 1). Does the roughness impact the CO₂ reduction current density? The authors must measure the electrochemically active surface area (ECSA) of the electrodes and compare ECSA-normalized activity to exclude the possible effect of surface roughness.

➤ **Response:**

We measured the electrochemically active surface area (ECSA) of CuNWs and compared ECSA-normalized activity of gaseous CO₂RR products. The ECSA was characterized by calculation of electrochemical double-layer capacitance (C_{dl}) over specific capacitance (C_s) (**Fig. R16**). AA-CuNW exhibited the largest ECSA (943.63 cm²) due to morphological changes through surface Cu oxidation during the synthesis (Supplementary Fig. 5). In addition, G-CuNW (744.54 cm²) and cAA-CuNW (600.00 cm²) show large ECSA compared to p-CuNW (329.54 cm²) due to the rough surface caused by GQDs, as confirmed in the TEM images (Fig. 1d–j). Based on ECSA of the electrodes, we calculated ECSA-normalized partial current densities in terms of H₂, CO, and C₂H₄ (**Fig. R17**). Partial current densities of CuNWs showed a similar trend with those of CuNWs before ECSA normalization (Fig. 3f and Supplementary Fig. 29). cAA-CuNW exhibited the highest CO and C₂H₄ productivity compared to others. This reveals that the highest $J_{C_2H_4}$ of cAA-CuNW is confirmed after considering the effect of surface roughness and nanoconfined AA promotes the intrinsic CO₂RR activity of cAA-CuNW.

➤ **Modification:**

(**Manuscript page 12 line 273-277**) We add explanation about the comparison of the ECSA-normalized partial current density for CuNW, G-CuNW, AA-CuNW, and cAA-CuNW.

(**Supplementary information page 33-34**) Fig. R16-R17 has been added as Supplementary Fig. 30-31. Experimental details were described.

Fig. R16 | ECSA measurement for CuNWs. CV plot of **a**, p-CuNW, **b**, G-CuNW, **c**, AA-CuNW, and **d**, cAA-CuNW in 1 M KOH electrolyte. **e**, Electrochemical double-layer capacitance (C_{dl}) and **f**, the calculated ECSA for CuNWs.

Fig. R17 | Comparison of ECSA-normalized CO₂RR productivities between CuNWs. ECSA-normalized partial current densities versus potentials of p-CuNW, G-CuNW, AA-CuNW, and cAA-CuNW were compared in terms of **a**, H₂, **b**, CO, and **c**, C₂H₄.

5. The authors pointed out that “AA can react with CO₂ and be oxidized to dehydroascorbic acid (DHA) with proton and electron donation”. It is unclear that whether AA directly participates in the CO₂ reduction reaction, or just serves as an electron or proton shuttle?

➤ **Response:**

As we compared the CV of AA and nanoconfined AA under N₂ and CO₂ supply (**Fig. R10**), intact AA did not show redox reversibility (**Fig. R10a**). However, nanoconfined AA showed the paired reduction peak (−0.36 V vs Ag/AgCl), ensuring enhanced redox reversibility under CO₂ atmosphere (**Fig. R10b**). In addition, the shifted oxidation peak of nanoconfined AA in CO₂ atmosphere (**Fig. R10c-d**) is due to the additional energy required to break the interaction between CO₂ and nanoconfined AA,^{33,34} considering that AA formed a strong hydrogen bonding in DFT simulation. Since the increased *CO coverage in cAA-CuNW was observed by *in situ* Raman spectroscopy (**Fig. R12a-d**), we think that reversible AA/DHA redox promotes CO₂-to-*CO conversion by enhancing electron and proton transfer during CO₂RR.

Fig. R11 presents schematic diagram for electrochemical AA/DHA redox cycle and related CO₂ capture. This contributes to CO₂-to-*CO conversion for CO₂RR. AA undergoes oxidation to generate electrons and protons, which are subsequently donated to CO₂, facilitating *CO formation on the catalyst. At the same time, DHA is reduced by applied potential from the cathode, thereby enabling a reversible redox cycle. Consequently, we think that AA does not exhibit intrinsic catalytic behavior and is not an active site. But, AA participates in CO₂RR by contributing the electrons and protons for CO₂-to-*CO conversion with the reversible AA/DHA redox.

➤ **Modification:**

(Manuscript page 9 line 200-206) We add explanation about the electrochemical analysis of AA and nanoconfined AA in N₂ and CO₂ atmospheres.

(Supplementary information page 19-20) Fig. R10-R11 has been added as Supplementary Fig. 16-17.

6. What is the mechanism of the nanoconfinement by graphene quantum dots, just physically covering the surface so that the AA cannot diffuse away? What adsorption or interaction do the AA molecules have with the graphene or the Cu nanowire surface, physisorption, chemisorption, or others?

➤ **Response:**

To understand the mechanism of interaction between AA, GQD, and Cu surface, we proposed a schematic illustration of the interaction among them (**Fig. R18**). Functional groups of GQDs such as hydroxyl and carboxyl group can be interacted and chelated to the nanometer-scale native oxide layer on the Cu surface.^{35,36} In the reaction between GQD and AA, AA removes impure oxygenated functional group in GQD through nucleophilic substitution (S_N2) reaction and reduces GQD through thermal elimination.^{37,38} Since the reaction between AA and GQD proceeded for 1 h at 95°C, most intermediates formed during the S_N2 reaction were eliminated. As a result, chemisorption between AA and GQD is not a preferentially considered interaction.

Reduced GQDs can be combined with AA *via* physisorption, such as π interaction at the basal plane or hydrogen bonding with edge functional groups. Molecules with endiol groups (e.g. catechol), carboxyl group, or aromatic rings can form 2-dimensional supramolecular systems with reduced graphene through π interaction or hydrogen bonding.³⁹⁻⁴² Furthermore, graphene reduced by AA exhibits excellent performance as an impervious barrier or protective coating agent due to little structural damage to the graphene during reduction,⁴³⁻⁴⁵ providing a confined system for the reversible redox of AA.

Immobilization of nanoconfined AA by Nafion also contributes to prevent AA dissolution. Nafion layer is impermeable to AA⁴⁶ and have been used to immobilize biomolecules in combination with graphene oxide.⁴⁷ Therefore, nanoconfined AA by GQDs can act as heterogeneous immobilized modulators that promote CO₂RR near Cu surface.

➤ **Modification:**

(Manuscript page 5-6 line 117-119) We add comments about the interaction between AA, GQDs, and Cu surface.

(Supplementary information page 4) Fig. R18 has been added as Supplementary Fig. 1. Detailed explanation about the interaction between AA, GQDs, and Cu surface was described.

(Supplementary information page 55) We added references for the explanation about the interaction between AA, GQDs, and Cu surface.

Fig. R18 | Schematic illustration of the interaction between AA, GQDs, and Cu surface.

7. *Stability of the nanoconfined AA during long-term electrolysis. As the authors indicated, “AA is easily dissolved into the electrolyte due to its high solubility in aqueous solutions”. Figure 2g showed that there was still DHA extracted in the solution from cAA-CuNW even with Nafion ionomer coating after 60 min, so the AA/DHA was detaching slowly from the surface. Thus, how long can the effect of nanoconfined AA last? Long-term electrolysis test with the quantification of AA/DHA in the electrode after the test must be shown.*

➤ **Response:**

To confirm the stability of nanoconfined AA on GQDs in CO₂RR, long-term CO₂RR was conducted in a flow cell electrolyzer with 1 M KOH electrolyte (**Fig. R19**). We compared the duration of C₂H₄ production between p-CuNW and cAA-CuNW at a total current density of 300 mA/cm². C₂H₄ FE for p-CuNW decreased from 56.1% to 21.3% within 2 h. However, C₂H₄ FE for cAA-CuNW was maintained over 50% for 8 h, indicating that the nanoconfined AA was well immobilized and stably operated with continuous redox cycle.

After the long-term CO₂RR stability test (8 h) in the flow cell, the chemical state of the cAA-CuNW was measured by Fourier transform infrared spectroscopy (FT-IR) (**Fig. R20**). When we compared the FT-IR peaks before and after CO₂RR of cAA-CuNW, the peaks for C=C stretching vibration, C=C–O asymmetric stretching vibration of the enol-hydroxyl group, and C–O vibration in the functional groups of AA were identical. In addition, the peaks at 1,207 and 1,153 cm⁻¹ were observed, which correspond to asymmetric and symmetric CF₂ stretch mode in Nafion.⁴⁸ Therefore, similar FT-IR spectra of the GDE before and after the CO₂RR stability test confirmed that the nanoconfined AA on GQDs is stable in cAA-CuNW during CO₂RR.

Additionally, we tried to quantify the amount of the eluted AA/DHA by collecting and analyzing the catholyte (1 M KOH) after 1 h CO₂RR (total current density of 300 mA/cm²) by high-performance liquid chromatography (HPLC) (**Fig. R21**). There was no HPLC peak for DHA or AA in the catholyte. This indicates that the amount of eluted AA/DHA was lower than HPLC detection limit. Note that the amount of eluted AA/DHA from the electrode for the flow cell is lower than that of the electrode used in Fig. 2g because the surface area of the electrode is reduced from 4.5 cm × 4.5 cm to 1 cm × 1 cm. We think that the nanoconfined AA on GQDs was not dissolved in the electrolyte

significantly after long-term CO₂RR.

Furthermore, we investigated the stability of cAA-CuNW in zero-gap membrane electrode assembly (MEA) electrolyzer with 0.1 M KHCO₃ anolyte (**Fig. R22**). cAA-CuNW exhibited outstanding stability of C₂H₄ production for 168 h. CO₂ electrolysis was terminated when H₂ FE reached 20%. Excellent long-term stability of cAA-CuNW reveals that nanoconfined AA continuously promotes CO₂-to-*CO conversion and *CO dimerization to enhance CO₂RR productivity.

➤ **Modification:**

(Manuscript page 13 line 294-306) We add explanation about the stability of nanoconfined AA on GQDs and long-term CO₂RR tests in a flow cell and MEA electrolyzers.

(Supplementary information page 38-40) Fig. R19, R20, and R22 have been added as Supplementary Fig. 35, 36, and 37, respectively. Experimental details were described.

Fig. R19 | Comparison of CO₂RR stability between p-CuNW and cAA-CuNW in flow cell electrolyzer with 1 M KOH electrolyte. CO₂RR of p-CuNW and cAA-CuNW proceeded at a total current density of 300 mA/cm² by chronopotentiometry.

Fig. R20 | FT-IR spectra of cAA-CuNW before and after CO₂RR stability test in the flow cell. The peaks at 1,207 and 1,153 cm^{-1} corresponding to asymmetric and symmetric CF₂ stretch mode in Nafion, respectively.

Fig. R21 | Detection of the extracted AA/DHA during CO₂RR by HPLC analysis. Reference AA (blue)/DHA (green), 1 M KOH solution (red), and catholyte (black) after 1 h CO₂RR at 1 M KOH electrolyte were analyzed for the detection of the extracted AA/DHA from cAA-CuNW. There was no peak in the catholyte at retention time of AA/DHA detection.

Fig. R22 | Long-term C₂H₄ production of cAA-CuNW in a MEA electrolyzer with 0.1 M KHCO₃ electrolyte. CO₂RR of cAA-CuNW proceeded at a total current density of 150 mA/cm² by chronopotentiometry.

8. The DFT calculations did not consider the cations in the electrolyte. However, recent study demonstrated that CO₂ electroreduction will not occur on Cu surface without metal cations (10.1038/s41929-021-00655-5). The authors showed CO₂ reduction on Cu(111) surface without the presence of any cations. Does this result conflict with the literature (10.1038/s41929-021-00655-5)?

➤ **Response:**

In response to the reviewer's opinions, we conducted a comprehensive reevaluation of our DFT calculations in the presence of a K⁺ ion on Cu (100). In the updated calculations, we performed AIMD simulations using 25 water molecules and one K⁺ ion (*Nat. Catal.* **4**, 654-662 (2021), *Nat. Commun.* **13**, 5482 (2022))^{4,5} to construct the solid-liquid interface (*ACS Catal.* **12**, 11530-11540 (2022), *J. Phys. Chem. C* **126**, 7841-7848 (2022)).^{6,7} After confirming the system had equilibrated, we used the last snapshot of the AIMD simulations as the starting structural configuration. Note that the position of the metal cation is crucial, as its interaction with adsorbates depends on its displacement (*J. Am. Chem. Soc.* **145**, 1897-1905 (2023), *J. Am. Chem. Soc.* **145**, 19601-19610 (2023)),^{8,9} and we confirmed that the optimized position of K⁺ is approximately 6 Å above the surface for both Cu (100) and AA/Cu (100).

The updated results demonstrated that the introduction of AA facilitated the protonation of *CO₂ to form *COOH, thus increasing the *CO coverage on the surface. In addition, it lowered the activation barrier of the rate-determining *CO dimerization, leading to higher production of C₂₊ products (**Fig. R23**). Thus, the updated calculations confirmed the effect of AA on promoting CO₂RR.

➤ **Modification:**

(Manuscript page 24-25 line 551-565) We include the aforementioned details regarding the explicit solvation model in the Methods section.

(Manuscript page 33 line 754-761) We add references for the explicit solvation model.

(Manuscript page 40) We change Fig. 5 to computational modeling of CO₂RR in the presence of a K⁺ ion on Cu (100) and AA/Cu (100) (**Fig. R23**).

Fig. R23 | Computational Modeling of CO₂RR on Cu (100) and AA/Cu (100). **a**, The Gibbs free energy diagram of (left) CO₂-to-*CO conversion and (right) *CO dimerization at 0 V (vs RHE). White circles indicate the reaction pathway involving the deprotonation of AA. H₂O is the proton source otherwise. **b**, The atomic structure of (left) the initial and (right) the final states of *CO₂ protonation from AA on AA/Cu (100) ($\text{CO}_2^* + \text{H}_2\text{O} + \text{AA} + \text{e}^- \rightarrow \text{*COOH} + \text{H}_2\text{O} + \text{ASC}^-$). **c**, Charge density difference ($\Delta\rho$) of *OCCO adsorption on (left) AA/Cu (100) and (right) Cu (100). The yellow and blue area represent an electron accumulation and depletion with an isosurface level of $0.005 \text{ e}/\text{\AA}^3$, respectively. The charge density difference is calculated as $\rho_{\text{total}} - \rho_{\text{ads+surf}} - \rho_{\text{sol}}$, where ρ_{total} , $\rho_{\text{ads+surf}}$, and ρ_{sol} , correspond to charge densities of the total system, the catalyst surface with adsorbates and solvent layers, respectively.

9. The authors reported a maximum $J_{C_2H_4}$ of 539 mA/cm² at -1.55 V vs. RHE, which is 2.9-fold higher than the highest $J_{C_2H_4}$ of pristine CuNW with 184 mA/cm² at -1.39 V vs RHE. However, the two current densities were measured at different potentials, so the comparison is unfair. Similarly, Supplementary Table 1 shows a comparison of various Cu-based CO₂RR catalysts for high-current ethylene production. What is the overpotential for each case? Without the inclusion of applied potentials in the table, such comparison is unfair and even misleading.

➤ **Response:**

We confirmed that the maximum $J_{C_2H_4}$ of p-CuNW and cAA-CuNW were measured at different overpotentials. However, HER of p-CuNW increased at higher potentials and $J_{C_2H_4}$ rather decreased (Fig. 3f). When comparing $J_{C_2H_4}$ in the same voltage range of -1.76 V (vs RHE) between p-CuNW and cAA-CuNW, $J_{C_2H_4}$ of cAA-CuNW is 499.26 mA/cm², which is 8.4 times higher than that of p-CuNW (59.45 mA/cm²).

We summarized overpotentials at maximum $J_{C_2H_4}$ and iR correction percentages in Supplementary Table 1 to avoid misleading and to make a fair comparison. (Table R2). Regarding that our work was the lowest iR compensation of 80%, the overpotential at maximum $J_{C_2H_4}$ for cAA-CuNW was not significantly high compared to other Cu catalyst for high C₂H₄ production.

In addition, it should be considered that the concentrations of electrolytes are also different for each reference, as the concentrations of electrolytes is the main factor determining the operating potential range of the cell. We further investigated the CO₂RR performance of cAA-CuNW at lower cathodic potentials with a 2 M KOH electrolyte (Fig. R24). Although the selectivity tendency of CO₂RR for cAA-CuNW in 1 and 2 M KOH electrolytes were similar, the overpotential in 2 M KOH decreased dramatically ($J_{C_2H_4}$ of 453 mA/cm², C₂H₄ FE of 56.3% at -0.57 V vs RHE). Therefore, the overpotential of cAA-CuNW is not significantly higher than others.

➤ **Modification:**

(Manuscript page 12 line 277-280) We add a comment on the results of CO₂RR in 2 M KOH electrolyte.

(Supplementary information page 35) Fig. R24 have been added as Supplementary Fig. 32.

(Supplementary information page 51) We add the results of CO₂RR using cAA-CuNW in 2 M KOH electrolyte and data of the overpotential at maximum J_{C₂H₄} in Supplementary Table 1 to avoid misleading and to make a fair comparison.

Table R2 | Comparison of various Cu-based CO₂RR catalysts for high-current C₂H₄ production.

Catalyst	Cell type	Electrolyte	J (mA/cm ²)	J _{C₂H₄} (mA/cm ²)	C ₂ H ₄ FE (%)	V (vs RHE)	iR Correction (%)	Ref.
cAA-CuNW	Flow cell	1 M KOH	888	539	60.7	-1.55	80	This work
		2 M KOH	805	453	56.3	-0.57	80	
S-HKUST-1	Flow cell	1 M KOH	400	229	57.2	-1.32	85	49
Quasi-graphitic C shell on Cu	Flow cell	1 M KOH	400	284.4	71.1	-0.69	85	50
Nanoporous Cu	Flow cell	1 M KOH	653	252	38.6	-0.67	Non-iR corrected	51
Fluorinated Cu	Flow cell	0.75 M KOH	1,600	1040	65	-0.89	85	52
Cu NPs + Nafion on Cu	Flow cell	7 M KOH	1,550	930	60	-3.23	100	11
Cu(100) grown under CO ₂	Flow cell	7 M KOH	580	388	67	-0.71	90	53
Cu film on PTFE	Flow cell	3.5 M KOH + 5 M KI	720	473	66	-0.67	100	54
Porous Cu	MEA	Pure Water	900	420	46.6	3.54	.	55
CAL-modified Cu	Slim, low-resistance flow cell	1 M H ₃ PO ₄ + 3 M KCl	1,200	372	31	4.2	.	56
CuO NS	Flow cell	1 M KHCO ₃	700	231	33	.	.	57
Cu-12	Flow cell	1 M KHCO ₃	322	232	72	-0.83	100	1

Fig. R24 | Effect of KOH concentration on shifting the potential of high-rate C₂H₄ production of cAA-CuNW. a, Gaseous product FEs and total current densities for cAA-CuNW with 2 M KOH electrolyte. b, Comparison of J_{C₂H₄} versus potentials of cAA-CuNW according to the KOH concentration.

Reviewer #3

In this manuscript, Kim et al reported that ascorbic acid (AA) can promote the C-C coupling by using AA-graphene quantum dots decorated Cu nanowires (cAA-CuNW), which facilitates the CO₂ electroreduction to ethylene production with higher Faradaic efficiency and current density. By using in situ characterization, the authors try to understand the role of AA on Cu for the boosted mechanism. Overall, the idea that using organic molecules to activate/boost adsorption of CO₂ is not novel, this work provides some fresh and interesting findings and conclusions for the related fields. However, some fundamental grounds have not been well illustrated. Current manuscript is promising but very preliminary. Therefore, it should be better for this manuscript to do major revision, and then this work will be more influential.

➤ **Response:**

We sincerely appreciate your acknowledgement of this work. The most interesting finding of our study is that we found that the redox activity of molecular additives affects the high-rate CO₂-to-*CO conversion, contributing to high C₂H₄ productivity. Moreover, we demonstrate for the first time the exploitation of the nanoconfinement effect *via* GQDs to ensure reversible redox cycling of small molecule additives in aqueous electrolyte. In particular, we performed extensive additional studies to better elucidate the fundamental basis of the nanoconfined AA mechanism. Below your concerns are carefully addressed point by point.

1. Although the authors tried to explain the role of AA and GQDs for the CO₂ electroreduction, I still feel confused how these two species work in the system. The authors should clearly explain the behaviour of corresponding components. For example, AA could be oxidized even without purging of CO₂. Therefore, will the AA oxidized form (i.e., DHA) has boosting role for CO₂ reduction?

➤ **Response:**

GQDs reduce the solubility of AA in electrolyte and ensure reversible AA/DHA redox through nanoconfinement of AA. To clearly demonstrate the mechanism of interaction between AA, GQD, and Cu surface, we proposed a schematic illustration of the interaction among Cu, GQD, and AA (**Fig. R18**). In the reaction between GQD and AA, AA removes

impure oxygenated functional group in GQD through S_N2 reaction and reduces GQD through thermal elimination.^{37,38} In particular, graphene-based materials reduced by AA exhibits excellent properties as an impervious barrier or protective coating agent due to little structural damage during the reaction.⁴³⁻⁴⁵ Therefore, the reduced GQDs can provide a suitable confinement for reversible AA/DHA redox.

In addition, we investigated the electrochemical behavior of AA and nanoconfined AA on GQDs under N_2 and CO_2 atmospheres (**Fig. R10**). When AA was dissolved in the electrolyte, the oxidation of AA was observed at 0.23–0.26 V (vs Ag/AgCl) in both N_2 and CO_2 atmosphere. However, the paired reduction peak was not observed at the reverse scans, indicating that AA was irreversibly converted to DHA on the electrode (**Fig. R10a**). We expect this to limit the promotion of CO_2 -to- CO^* conversion; the enol group in AA was oxidized to carbonyl group, which does not donate electrons and protons to CO_2 . However, nanoconfined AA on GQDs shows the paired reduction peak (–0.36 V vs Ag/AgCl) in both N_2 and CO_2 atmosphere (**Fig. R10b-d**). Since the increased *CO coverage in cAA-CuNW was observed by *in situ* Raman spectroscopy (**Fig. R12a-d**), we think that reversible AA/DHA redox promotes CO_2 -to- *CO conversion by enhancing electron and proton transfer during CO_2RR .²⁴⁻²⁶ Based on these results, we present a schematic diagram of the electrochemical behavior of nanoconfined AA and CO_2RR in **Fig. R11**.

Finally, we investigated the CO_2RR activity of DHA-CuNW to prove the role of DHA in promoting CO_2RR (**Fig. R25**). DHA-CuNW showed low CO FE of 8.2%, while AA-CuNW showed over 2-fold increase in CO FE (17.0%) at –0.81 V (vs RHE). This suggests that CO_2RR was not promoted using DHA due to the absence of enol group that can donate electrons and protons.

➤ **Modification:**

(Manuscript page 5-6 line 117-119) We add a comment on the interaction between AA, GQDs, and Cu surface.

(Manuscript page 9 line 200-206) We add explanation about the electrochemical analysis of AA and nanoconfined AA on GQDs in N_2 and CO_2 atmospheres.

(Manuscript page 11 line 246-247) We add a comment on the comparison of the gaseous

product selectivity of p-CuNW, DHA-CuNW, and AA-CuNW.

(Supplementary information page 4) Fig. R18 has been added as Supplementary Fig. 1. Detailed explanation about the interaction between AA, GQDs, and Cu surface was described.

(Supplementary information page 19-20) Fig. R10-R11 has been added as Supplementary Fig. 16-17.

(Supplementary information page 30) Fig. R25 has been added as Supplementary Fig. 27.

(Supplementary information page 55) We add references for the explanation about the interaction between AA, GQDs, and Cu surface.

Fig. R25 | Effect of DHA on electrochemical CO₂RR performance of CuNW. Comparison of the gaseous product selectivity of p-CuNW, DHA-CuNW, and AA-CuNW.

2. The authors should provide more supportive and direct evidence to illustrate the capture of CO₂ with the help of AA. Some works from Lin Zhuang group (ACS Catalysis, 2022, 12, 1004-1011; ACS Energy Letter, 2022, 7, 4045-4051.) tried to utilize *in situ* FTIR to monitor the adsorption and activation of CO₂ molecules with the help of some organic molecules.

➤ **Response:**

The suggested reference presented *in situ* attenuated total reflectance surface-enhanced IR absorption spectroscopy (ATR-SERIAS) to monitor the adsorption and activation of CO₂ molecules by detecting *CO₂⁻.⁵⁸ They found that the *CO₂⁻ peak at 1,580 cm⁻¹ for bare Cu gradually disappears as the potential decreases, however, the peak rarely appears for Cu containing molecular additive that facilitates high-rate CO₂ conversion.

We further analyzed *in situ* Raman spectroscopy for monitoring *CO₂⁻ on the catalyst surface during CO₂RR to directly reveal the effect of AA on the capture of CO₂, (Fig. R26). The Raman spectra of both p-CuNW and cAA-CuNW exhibit a *CO₂⁻ peak in the region of 1,500-1,600 cm⁻¹, which corresponds to the asymmetric stretching vibration of *CO₂⁻.⁵⁹ This peak was observed at 0.2 V (vs RHE, non-iR corrected) and gradually disappeared as the potential decreased. Considering that the Cu-CO binding peaks at 200-400 cm⁻¹ increased as the potential decreased, the disappearance of *CO₂⁻ peak was attributed to subsequent *CO formation. In addition, we found that the *CO₂⁻ peak intensity of cAA-CuNW is lower than that of p-CuNW, which is contrary to the superior *CO peak intensity of cAA-CuNW. In accordance with the suggested ATR-SERIAS reports,⁵⁸ these Raman spectral differences are attributed to the promoted CO₂-to-*CO conversion of cAA-CuNW.

In addition, we also performed *in situ* ATR-SERIAS measurement to detect the reaction intermediates for p-CuNW and cAA-CuNW during CO₂RR (Fig. R27). The electrochemical measurements for *in situ* ATR-SEIRAS were carried out with a CO₂ saturated 1 M KOH electrolyte and a constant CO₂ flow of 20 sccm was maintained throughout the reduction procedure. The *in situ* ATR-SERIAS results of cAA-CuNW shows a *CO peak corresponding the vibration of *CO (ν(*CO)) at 1,820 cm⁻¹, while that of p-CuNW shows significantly low *CO peak intensity. Therefore, nanoconfined AA facilitates high-rate CO₂-to-*CO conversion and ensures higher *CO coverage on the catalyst surface.

➤ **Modification:**

(Manuscript page 14 line 323-326) We add explanation about the *in situ* Raman spectroscopy for monitoring $^*\text{CO}_2^-$ to directly reveal the effect of AA on the capture of CO_2 .

(Manuscript page 30 line 687-689) We add references for explanation about *in situ* Raman spectra for monitoring $^*\text{CO}_2^-$.

(Supplementary information page 42) Fig. R26 have been added as Supplementary Fig. 39.

Fig. R26 | Monitoring $^*CO_2^-$ on CuNWs during CO_2RR . *In situ* Raman spectra obtained during the CO_2RR under different applied potentials from **a**, p-CuNW, and **b**, cAA-CuNW region of 1,400-1,700 cm^{-1} .

Fig. R27 | *In situ* ATR-SEIRAS obtained during CO₂RR under different applied potentials from CuNWs. *In situ* ATR-SEIRAS results under different applied potentials from **a**, p-CuNW and **b**, cAA-CuNW region of 1,700-2,200 cm⁻¹. The electrochemical measurements for ATR-SEIRAS were carried out with a CO₂ saturated 1 M KOH electrolyte and a constant CO₂ flow of 20 sccm was maintained throughout the reduction procedure. The spectroelectrochemical cell was integrated into a FT-IR spectrophotometer (VERTEX 80v, Bruker) equipped with a mercury cadmium telluride detector and a variable angle specular reflectance accessory (VeemaxIII, Pike Technologies). All spectroscopic measurements were conducted at a 4 cm⁻¹ spectral resolution.

3. In the materials characterization part, what kind of TEM grid the authors were using? In fact, EDS is not accurate to explain the ratio of C/N/O/Cu, as most TEM grid can also false signals on EDS with the aforementioned elements. Similar situations also happen on XPS measurement. Therefore, the authors should carefully treat these data and results, and discuss them critically. Otherwise, it is suggested to use more accurate analytical methods to draw out the conclusion about non-metal distribution around Cu. Also, according to the TEM images (Fig S4), the outer-shell (3~5nm) is almost CuO, which is contradictory to the discussion that O barely detected.

➤ **Response:**

We utilized lacey formvar carbon-coated square Au grid and Be cradle holder for TEM analysis (Fig. R28) and P type Boron doped Si wafer for XPS analysis to minimize unwanted X-ray signals. We appreciate the reviewer for this comment, and we added the detailed sampling information in experimental section to clearly indicate the reliability of the X-ray elemental analysis.

The TEM images in Supplementary Fig. 5 (Supplementary Fig. 4 for the original manuscript) show nanomorphology of AA-CuNW and oxidation of the outer shell was observed as noted by the reviewer. The Cu surface of AA-CuNW reacted with oxygen in organic solvents due to the absence of GQDs, leading to partial oxidation during surface functionalization. TEM EDS analysis (Fig. 2a) showed that the fraction of O for AA-CuNW (15.8%) is much higher than that of G-CuNW (0.4%), which is consistent with the analysis in Supplementary Fig. 5.

➤ **Modification:**

(Manuscript page 20 line 456-457 and 459-460) We add the detailed sampling information about TEM and XPS analysis in experimental section to clearly indicate the reliability of the X-ray elemental analysis.

Fig. R28 | A photograph of **a**, Be cradle TEM holder with **b**, lacey formvar carbon-coated square Au grid.

4. For the catalysis part, the authors should provide more post-catalysis characterizations including TEM and XRD to support the discussion the authors mentioned. It is also recommended that the authors check all catalytic performance data, as some error bars are >10% (relative error). Therefore, these data may not be very convincing to present the trend and differences between all samples.

➤ **Response:**

We conducted characterizations after CO₂RR including XRD, SEM, TEM, and TEM EDS to investigate operation stability of CuNWs for high C₂H₄ production. The XRD patterns of all CuNWs after CO₂RR exhibited Cu₂O (111) peaks due to surface oxidation by electrolyte (**Fig. R29**). However, the 1-dimensional structures of CuNWs were maintained as shown in SEM images (**Fig. R30**), indicating that there was no significant structural transformation during CO₂RR. The crystal structure and atomic distribution of CuNWs were analyzed using TEM and TEM EDS. CuNWs exhibited a rough Cu₂O surface after CO₂RR (**Fig. R31**), consistent with the XRD analysis results. In addition, the lattice of GQDs and amorphous nanostructure of AA was still observed at the outer shell of G-CuNW, AA-CuNW, and cAA-CuNW, suggesting that each material was well immobilized on the CuNW surface during CO₂RR. The elemental distribution confirmed that the uniform distribution of Cu, C, and O atoms on the entire surface of the CuNW structure was maintained after CO₂RR (**Fig. R32**).

The error bars for CO₂RR data (gas and liquid products) represent standard deviation based on three independent samples. Gas product GC data per each sample were collected as the average value measured at 5, 30, and 55 min intervals. Therefore, the error increases when the performance fluctuation of the catalyst is severe for each measurement interval. C₂H₄ FE of CuNW drops more than 10% in 1 h at 300 mA/cm² (**Fig. R19**), implying that there can be considerable variation in each measurement. Therefore, the error values for our experiments are comparable to those in various literatures on CO₂RR.^{56,60,61}

➤ **Modification:**

(**Manuscript page 8 line 176-188**) We add explanation about the characterizations of CuNWs after CO₂RR to prove the reliable operation of nanoconfined AA for high C₂H₄

production.

(Manuscript page 20-21 line 460-463) We add the detailed sampling information about characterizations of CuNWs after CO₂RR in experimental section.

(Manuscript page 22 line 506-509) We add explanation about the data collection and calculation of the error bars for CO₂RR data in experimental section.

(Supplementary information page 14-17) Fig. R29-R32 have been added as Supplementary Fig. 11-14.

Fig. R29 | Crystal structure analysis of CuNWs after CO₂RR. a, XRD patterns of p-CuNW, G-CuNW, AA-CuNW, and cAA-CuNW on PTFE substrate before CO₂RR. **b**, Magnified XRD patterns of each sample before CO₂RR. **c**, XRD patterns of p-CuNW, G-CuNW, AA-CuNW, and cAA-CuNW on PTFE substrate after CO₂RR. **d**, Magnified XRD patterns of each sample after CO₂RR.

Fig. R30 | Investigation of microstructures of CuNWs after CO₂RR. SEM (top) and HR-SEM (bottom) images of **a–b**, p-CuNW, **c–d**, G-CuNW, **e–f**, AA-CuNW, and **g–h**, cAA-CuNW after CO₂RR.

Fig. R31 | Investigation of surface structures of CuNWs after CO₂RR. TEM (top) and HR-TEM (bottom) images of **a–b**, p-CuNW, **c–d**, G-CuNW, **e–f**, AA-CuNW, and **g–h**, cAA-CuNW after CO₂RR.

Fig. R32 | Investigation of elemental distribution of CuNWs after CO₂RR. TEM EDS mapping of Cu, C, O, and the combination of Cu-C-O for **a–d**, p-CuNW, **e–h**, G-CuNW, **i–l**, AA-CuNW, and **m–p**, cAA-CuNW after CO₂RR, respectively.

5. It is good that the authors utilized *in situ* Raman to quantitatively compare the ratio of $C=O_{bridge}$ and $C=O_{top}$ peaks and correlate it to the CO coverage. However, according to Fig. 3, the best ethylene production potential lies below $-0.8V$ vs RHE. Therefore, why did the author stop the *in situ* Raman testing at $-0.8V$ vs RHE? It is suggested to collect all spectra alongside the shifting of the biased potential, and then find the trend.

➤ **Response:**

We appreciate the reviewer for indicating potential differences between CO₂RR and *in situ* Raman spectroscopy. The highest C₂H₄ production potential of cAA-CuNW was -1.55 V (vs RHE), which is much lower than the potential from *in situ* Raman spectroscopy (-0.8 V vs RHE, non-iR corrected). However, it is difficult to measure *in situ* Raman spectroscopy at potentials below -0.8 V (vs RHE, non-iR corrected) due to laser focusing interference induced by gaseous products.⁶² Fig. R33 shows the focusing interference of immersive lens during measurement of *in situ* Raman spectroscopy up to -0.9 V (vs RHE, non-iR corrected). Due to the focusing interferences, most literatures on CO₂RR studies utilizing *in situ* Raman spectroscopy did not measure the spectra at low potentials, even though the optimal overpotential for product production is lower (Fig. R34).⁶³⁻⁶⁵ Therefore, addressing the focusing interference of the immersive lens by gaseous products is a critical challenge that must be overcome for future *in situ* Raman spectroscopy. Structural improvements in flow cell electrolyzers for elimination of gas bubble can be a promising approach to solving the focusing problem. We anticipate that developing an electrolyzer structure in future work will allow us to perform *in situ* Raman spectroscopy even at low potentials.

➤ **Modification:**

(Manuscript page 21 line 478-479) We add explanation about potential limitations of *in situ* Raman spectroscopy due to laser focusing interference of gaseous products.

Fig. R33 | Optical microscopy images of p-CuNW and cAA-CuNW during measurement of *in situ* Raman spectroscopy with different potentials.

Fig. R34 | *In situ* Raman spectroscopy in previous reports. Potentials for CO₂RR (left) and *in situ* Raman spectroscopy (right) reproduced from **a–b**, X. Chen et al., *Nat. Catal.* **4**, 20–27 (2021),⁶³ **c–d**, Y. Jiang et al., *Adv. Sci.* **9**, 2105292 (2022),⁶⁵ and **e–f**, C. Y. J. Lim et al., *Nat. Commun.* **14**, 335 (2023).⁶⁴

6. For the DFT calculations, please explain why Cu(111) was utilized for modelling? According to the Fig. 1, TEM images exhibited the Cu (100). Multifacets of Cu could also be found in Fig. S4-S5.

➤ **Response:**

We appreciate the reviewer's comment. There was a discrepancy between the most common CuNW facet (Cu(100)) and the modeling of Cu (111) in the original manuscript.

In response to the reviewer's opinions, we conducted a comprehensive reevaluation of our DFT calculations on Cu (100). In addition, we included the effect of K⁺ ion in the explicit solvation model. In the updated calculations, we performed AIMD simulations using 25 water molecules and one K⁺ ion (*Nat. Catal.* **4**, 654-662 (2021), *Nat. Commun.* **13**, 5482 (2022))^{4,5} to construct the solid-liquid interface (*ACS Catal.* **12**, 11530-11540 (2022), *J. Phys. Chem. C* **126**, 7841-7848 (2022)).^{6,7} After confirming the system had equilibrated, we used the last snapshot of the AIMD simulations as the starting structural configuration. Note that the position of the metal cation is crucial, as its interaction with adsorbates depends on its displacement (*J. Am. Chem. Soc.* **145**, 1897-1905 (2023), *J. Am. Chem. Soc.* **145**, 19601-19610 (2023)),^{8,9} and we confirmed that the optimized position of K⁺ is approximately 6 Å above the surface for both Cu (100) and AA/Cu (100).

The updated results demonstrated that the introduction of AA facilitated the protonation of *CO₂ to form *COOH, thus increasing the *CO coverage on the surface. In addition, it lowered the activation barrier of the rate-determining *CO dimerization, leading to higher production of C₂₊ products (**Fig. R23**). Thus, the updated calculations confirmed the effect of AA on promoting CO₂RR.

➤ **Modification:**

(Manuscript page 24-25 line 551-565) We include the aforementioned details regarding the explicit solvation model in the Methods section.

(Manuscript page 33 line 754-761) We add references for the explicit solvation model.

(Manuscript page 40) We change Fig. 5 to computational Modeling of CO₂RR in the presence of a K⁺ ion on Cu (100) and AA/Cu (100) (**Fig. R23**).

7. The authors should provide the whole pathway gibbs free energy diagram for CO₂ reduction towards CO and ethylene respectively, rather than providing some data on CO adsorption and OCCO. Also, HER should also be discussed. For the last paragraph of result and discussion (Considering that the activation xxxxx), there is no systematic data in this manuscript to support that conclusion.

➤ **Response:**

We appreciate the reviewer's comment and acknowledge the importance of investigating the entire pathway to compare the catalytic properties with and without AA. However, we would highlight that a complete understanding of CO₂RR pathway is still evolving⁶⁶ and the RDS for C₂H₄ production has been consistently reported as *CO dimerization.^{67,68} Thus, we chose to focus on the *CO formation and dimerization in our study.

Regarding the second comment, we examined the effect of AA on the competitive HER (**Fig. R35**). The energetics of the first protonation step to form the adsorbed *H were found to be less favorable compared to *CO₂ adsorption, both with and without AA. Although the introduction of AA lowered the energy barrier of *H formation, the second protonation step remained unfavorable compared to *CO₂ adsorption. This confirms that introducing AA does not increase the catalytic activity of HER.

In situ Raman spectroscopy revealed that the surface *CO coverage was unrivaled on cAA-CuNW compared to other CuNWs (**Fig. R12**). This high *CO coverage of cAA-CuNW is due to the continuous CO₂-to*CO conversion *via* proton and electron transfer from the redox of nanoconfined AA. We also found that *CO dimerization was activated in cAA-CuNW by comparing *CO binding modes of CuNWs. cAA-CuNW show an appropriate CO_{bridge}/CO_{atop} ratio even when the potential increases up to -0.8 V (vs RHE), while p-CuNW and G-CuNW show an excessive CO_{atop} or CO_{bridge} as the potential increases (**Fig. R13**). We also compared C₂H₄ FE and CH₄ FE to confirm whether the *CO on catalyst surface is favorable to dimerization or not. (**Fig. R9**). We confirmed that C₂H₄ FE/CH₄ FE was much higher for cAA-CuNW, indicating that *CO on the cAA-CuNW prefers dimerization.

Since all the key reaction steps of CO₂RR (*CO₂ protonation, *CO dimerization, HER) were systematically compared with and without AA, we expect our GC-DFT and

experimental results support the conclusions of the experiments.

➤ **Modification:**

(Manuscript page 18 line 404-408) We add explanation about the effect of AA on the competitive HER.

(Manuscript page 18-19 line 409-412 and 431-433) We revise the conclusion of DFT calculation in accordance with the additional investigation.

(Manuscript page 39) We update graphs of *in situ* Raman spectroscopy of G-CuNW and AA-CuNW (Fig. R12) in Fig. 4c-f.

(Manuscript page 39) We update graphs of the comparison of integral area ratios of CO_{atop} and CO_{bridge} between CuNWs (Fig. R13) in Fig. 4i.

(Supplementary information page 31) Fig. R9 has been added as Supplementary Fig. 28.

(Supplementary information page 48) Fig. R35 has been added as Supplementary Fig. 45.

Fig. R35 | The Gibbs free energy diagram of the HER. The reaction pathway involving the deprotonation of AA is highlighted with white circles, while H₂O is the proton source otherwise.

Minor comments:

1. *Some symbols need to be checked. Please keep the united expressions for the whole manuscript.*

➤ **Response:**

We appreciate the reviewer for this comment. We checked the symbols used in the manuscript and united their expressions.

➤ **Modification:**

We revise the expression of the catalyst to keep the united expressions for the whole manuscript.

(Manuscript page 37) We unite the expression of the catalyst and the color of symbols in Figs. 2d–e as in other graphs.

(Manuscript page 38) We unite the color of symbols in Fig. 3f as in other graphs.

2. *The introduction and organization of the whole manuscript should be rearranged.*

➤ **Response:**

In accordance with reviewer's comment, we rearranged the introduction and organization of the manuscript.

➤ **Modification:**

(Manuscript page 2-4 line 24-25, 31-33, 55-72, and 93-94) We revise the abstract and introduction of the manuscript to highlight the novelty of the role of nanoconfined AA for CO₂-C₂H₄ conversion.

(Manuscript page 4 line 73-81) We rearrange the introduction of the manuscript to correct the logical flow of a paragraph.

References

1. Li, F. et al. Molecular tuning of CO₂-to-ethylene conversion. *Nature* **577**, 509-513 (2020).
2. Wang, X. et al. Efficient electrically powered CO₂-to-ethanol *via* suppression of deoxygenation. *Nat. Energy* **5**, 478-486 (2020).
3. Patel, A. M., Vijay, S., Kastlunger, G., Nørskov, J. K. & Chan, K. Generalizable trends in electrochemical protonation barriers. *J. Phys. Chem. Lett.* **12**, 5193-5200 (2021).
4. Monteiro, M. C. O. et al. Absence of CO₂ electroreduction on copper, gold and silver electrodes without metal cations in solution. *Nat. Catal.* **4**, 654-662 (2021).
5. Shin, S.-J. et al. A unifying mechanism for cation effect modulating C₁ and C₂ productions from CO₂ electroreduction. *Nat. Commun.* **13**, 5482 (2022).
6. Qian, S.-J. et al. Critical role of explicit inclusion of solvent and electrode potential in the electrochemical description of nitrogen reduction. *ACS Catal.* **12**, 11530-11540 (2022).
7. Yan, H.-M., Wang, Z.-X., Wang, Y.-M., Xia, G.-J. & Wang, Y.-G. Fast transformation of CO₂ into CO *via* a hydrogen bond network on the Cu electrocatalysts. *J. Phys. Chem. C* **126**, 7841-7848 (2022).
8. Qin, X., Vegge, T. & Hansen, H. A. Cation-coordinated inner-sphere CO₂ electroreduction at Au–water interfaces. *J. Am. Chem. Soc.* **145**, 1897-1905 (2023).
9. Ye, C., Dattila, F., Chen, X., López, N. & Koper, M. T. M. Influence of cations on HCOOH and CO formation during CO₂ reduction on a Pd_{ML}Pt(111) electrode. *J. Am. Chem. Soc.* **145**, 19601-19610 (2023).
10. Zhang, T. et al. Regulation of functional groups on graphene quantum dots directs selective CO₂ to CH₄ conversion. *Nat. Commun.* **12**, 5265 (2021).

11. García de Arquer, F. P. et al. CO₂ electrolysis to multicarbon products at activities greater than 1 A cm⁻². *Science* **367**, 661-666 (2020).
12. Wang, J. et al. Selective CO₂ electrochemical reduction enabled by a tricomponent copolymer modifier on a copper surface. *J. Am. Chem. Soc.* **143**, 2857-2865 (2021).
13. Wei, X. et al. Highly selective reduction of CO₂ to C₂₊ hydrocarbons at copper/polyaniline interfaces. *ACS Catal.* **10**, 4103-4111 (2020).
14. Liang, H.-Q. et al. Hydrophobic copper interfaces boost electroreduction of carbon dioxide to ethylene in water. *ACS Catal.* **11**, 958-966 (2021).
15. Wakerley, D. et al. Bio-inspired hydrophobicity promotes CO₂ reduction on a Cu surface. *Nat Mater.* **18**, 1222-1227 (2019).
16. Lin, Y. et al. Tunable CO₂ electroreduction to ethanol and ethylene with controllable interfacial wettability. *Nat. Commun.* **14**, 3575 (2023).
17. Han, Z. et al. CO₂ reduction selective for C_{≥2} products on polycrystalline copper with N-substituted pyridinium additives. *ACS Cent. Sci.* **3**, 853-859 (2017).
18. Ahn, S. et al. Poly-amide modified copper foam electrodes for enhanced electrochemical reduction of carbon dioxide. *ACS Catal.* **8**, 4132-4142 (2018).
19. Xie, M. S. et al. Amino acid modified copper electrodes for the enhanced selective electroreduction of carbon dioxide towards hydrocarbons. *Energy Environ. Sci.* **9**, 1687-1695 (2016).
20. Prajapati, A. et al. Fully-integrated electrochemical system that captures CO₂ from flue gas to produce value-added chemicals at ambient conditions. *Energy Environ. Sci.* **15**, 5105-5117 (2022).
21. Cheng, Y., Hou, J. & Kang, P. Integrated capture and electroreduction of flue gas CO₂ to formate using amine functionalized SnO_x nanoparticles. *ACS Energy Lett.* **6**, 3352-

- 3358 (2021).
22. Kim, D. et al. Electrocatalytic reduction of low concentrations of CO₂ gas in a membrane electrode assembly electrolyzer. *ACS Energy Lett.* **6**, 3488-3495 (2021).
 23. Zhang, B. et al. Steering CO₂ electroreduction toward methane or ethylene production. *Nano Energy* **88**, 106239 (2021).
 24. Cardoso, J. C. et al. The effective role of ascorbic acid in the photoelectrocatalytic reduction of CO₂ preconcentrated on TiO₂ nanotubes modified by ZIF-8. *J. Electroanal. Chem.* **856**, 113384 (2020).
 25. Pastero, L. et al. CO₂ capture and sequestration in stable Ca-oxalate, via Ca-ascorbate promoted green reaction. *Sci. Total Environ.* **666**, 1232-1244 (2019).
 26. Pastero, L., Marengo, A., Boero, R. & Pavese, A. Non-conventional CO₂ sequestration via vitamin C promoted green reaction: Yield evaluation. *J. CO₂ Util.* **44**, 101420 (2021).
 27. Zhu, H. et al. Coupling of graphene quantum dots with MnO₂ nanosheets for boosting capacitive storage in ionic liquid electrolyte. *Chem. Eng. J.* **437**, 135301 (2022).
 28. Tang, D. et al. Carbon quantum dot/NiFe layered double-hydroxide composite as a highly efficient electrocatalyst for water oxidation. *ACS Appl. Mater. Interfaces* **6**, 7918-7925 (2014).
 29. Jin, H. et al. Graphene quantum dots supported by graphene nanoribbons with ultrahigh electrocatalytic performance for oxygen reduction. *J. Am. Chem. Soc.* **137**, 7588-7591 (2015).
 30. Sun, C. & Bai, B. Gas diffusion on graphene surfaces. *Phys. Chem. Chem. Phys.* **19**, 3894-3902 (2017).
 31. Torrisi, L., Cutroneo, M., Torrisi, A. & Silipigni, L. Nitrogen diffusion in graphene

- oxide and reduced graphene oxide foils. *Vacuum* **194**, 110632 (2021).
32. Cho, H.-H., Yang, H., Kang, D. J. & Kim, B. J. Surface engineering of graphene quantum dots and their applications as efficient surfactants. *ACS Appl. Mater. Interfaces* **7**, 8615-8621 (2015).
 33. Choi, G. H. et al. Electrochemical direct CO₂ capture technology using redox-active organic molecules to achieve carbon-neutrality. *Nano Energy* **112**, 108512 (2023).
 34. Iijima, G. et al. Mechanism of CO₂ capture and release on redox-active organic electrodes. *Energy Fuels* **37**, 2164-2177 (2023).
 35. Dou, L. et al. Solution-processed copper/reduced-graphene-oxide core/shell nanowire transparent conductors. *ACS Nano* **10**, 2600-2606 (2016).
 36. Huang, S. et al. High-performance suspended particle devices based on copper-reduced graphene oxide core-shell nanowire electrodes. *Adv. Energy Mater.* **8**, 1703658 (2018).
 37. Chua, C. K. & Pumera, M. Chemical reduction of graphene oxide: A synthetic chemistry viewpoint. *Chem. Soc. Rev.* **43**, 291-312 (2014).
 38. Agarwal, V. & Zetterlund, P. B. Strategies for reduction of graphene oxide—A comprehensive review. *Chem. Eng. J.* **405**, 127018 (2021).
 39. Georgakilas, V. et al. Noncovalent functionalization of graphene and graphene oxide for energy materials, biosensing, catalytic, and biomedical applications. *Chem. Rev.* **116**, 5464-5519 (2016).
 40. Yang, M., Hou, Y. & Kotov, N. A. Graphene-based multilayers: Critical evaluation of materials assembly techniques. *Nano Today* **7**, 430-447 (2012).
 41. Griessl, S. et al. Self-assembled two-dimensional molecular host-guest architectures from trimesic acid. *Single Mol.* **3**, 25-31 (2002).
 42. Wang, Y., Shi, Z. & Yin, J. Facile synthesis of soluble graphene *via* a green reduction

- of graphene oxide in tea solution and its biocomposites. *ACS Appl. Mater. Interfaces* **3**, 1127-1133 (2011).
43. Su, Y. et al. Impermeable barrier films and protective coatings based on reduced graphene oxide. *Nat. Commun.* **5**, 4843 (2014).
 44. Moon, I. K., Lee, J., Ruoff, R. S. & Lee, H. Reduced graphene oxide by chemical graphitization. *Nat. Commun.* **1**, 73 (2010).
 45. Fernández-Merino, M. J. et al. Vitamin C is an ideal substitute for hydrazine in the reduction of graphene oxide suspensions. *J. Phys. Chem. C* **114**, 6426-6432 (2010).
 46. Gerhardt, G. A. et al. Nafion-coated electrodes with high selectivity for CNS electrochemistry. *Brain Res.* **290**, 390-395 (1984).
 47. Chen, G., Sun, H. & Hou, S. Electrochemistry and electrocatalysis of myoglobin immobilized in sulfonated graphene oxide and Nafion films. *Anal. Biochem.* **502**, 43-49 (2016).
 48. Kunimatsu, K. et al. *In situ* ATR-FTIR study of oxygen reduction at the Pt/Nafion interface. *Phys. Chem. Chem. Phys.* **12**, 621-629 (2010).
 49. Wen, C. F. et al. Highly ethylene-selective electrocatalytic CO₂ reduction enabled by isolated Cu–S motifs in metal–organic framework based precatalysts. *Angew. Chem. Int. Ed.* **61**, e202111700 (2022).
 50. Kim, J.-Y. et al. Quasi-graphitic carbon shell-induced Cu confinement promotes electrocatalytic CO₂ reduction toward C₂₊ products. *Nat. Commun.* **12**, 3765 (2021).
 51. Lv, J.-J. et al. A highly porous copper electrocatalyst for carbon dioxide reduction. *Adv. Mater.* **30**, 1803111 (2018).
 52. Ma, W. et al. Electrocatalytic reduction of CO₂ to ethylene and ethanol through hydrogen-assisted C–C coupling over fluorine-modified copper. *Nat. Catal.* **3**, 478-487

- (2020).
53. Wang, Y. et al. Catalyst synthesis under CO₂ electroreduction favours faceting and promotes renewable fuels electrosynthesis. *Nat. Catal.* **3**, 98-106 (2020).
 54. Dinh, C.-T. et al. CO₂ electroreduction to ethylene *via* hydroxide-mediated copper catalysis at an abrupt interface. *Science* **360**, 783-787 (2018).
 55. Li, W. et al. Bifunctional ionomers for efficient co-electrolysis of CO₂ and pure water towards ethylene production at industrial-scale current densities. *Nat. Energy* **7**, 835-843 (2022).
 56. Huang, J. E. et al. CO₂ electrolysis to multicarbon products in strong acid. *Science* **372**, 1074-1078 (2021).
 57. Wang, X. et al. Morphology and mechanism of highly selective Cu(ii) oxide nanosheet catalysts for carbon dioxide electroreduction. *Nat. Commun.* **12**, 794 (2021).
 58. Li, J. et al. Polyquinone modification promotes CO₂ activation and conversion to C₂₊ products over copper electrode. *ACS Energy Lett.* **7**, 4045-4051 (2022).
 59. Zhao, Y. et al. Elucidating electrochemical CO₂ reduction reaction processes on Cu(*hkl*) single-crystal surfaces by *in situ* Raman spectroscopy. *Energy Environ. Sci.* **15**, 3968-3977 (2022).
 60. Zhong, M. et al. Accelerated discovery of CO₂ electrocatalysts using active machine learning. *Nature* **581**, 178-183 (2020).
 61. Xu, H. et al. Highly selective electrocatalytic CO₂ reduction to ethanol by metallic clusters dynamically formed from atomically dispersed copper. *Nat. Energy* **5**, 623-632 (2020).
 62. Deng, Y. & Yeo, B. S. Characterization of electrocatalytic water splitting and CO₂ reduction reactions using *in situ/operando* Raman spectroscopy. *ACS Catal.* **7**, 7873-

- 7889 (2017).
63. Chen, X. et al. Electrochemical CO₂-to-ethylene conversion on polyamine-incorporated Cu electrodes. *Nat. Catal.* **4**, 20-27 (2021).
 64. Lim, C. Y. J. et al. Surface charge as activity descriptors for electrochemical CO₂ reduction to multi-carbon products on organic-functionalised Cu. *Nat. Commun.* **14**, 335 (2023).
 65. Jiang, Y. et al. Structural reconstruction of Cu₂O superparticles toward electrocatalytic CO₂ reduction with high C₂₊ products selectivity. *Adv. Sci.* **9**, 2105292 (2022).
 66. Nitopi, S. et al. Progress and perspectives of electrochemical CO₂ reduction on copper in aqueous electrolyte. *Chem. Rev.* **119**, 7610-7672 (2019).
 67. Garza, A. J., Bell, A. T. & Head-Gordon, M. Mechanism of CO₂ reduction at copper surfaces: Pathways to C₂ products. *ACS Catal.* **8**, 1490-1499 (2018).
 68. Li, Z., Zhang, T., Raj, J., Roy, S. & Wu, J. Revisiting reaction kinetics of CO electroreduction to C₂₊ products in a flow electrolyzer. *Energy Fuels* **37**, 7904-7910 (2023).

REVIEWER COMMENTS

Reviewer #1 (Remarks to the Author):

The authors have made revisions and addressed my comments, and the quality of the work has been highly improved. I recommend the publication.

Reviewer #2 (Remarks to the Author):

The authors have made substantial efforts to address the previously raised comments in a satisfactory manner. After the revision, the quality of the manuscript has been greatly improved with the new data and discussions. Particularly, the role and underlying mechanisms of the nanoconfined AA for improving the CO₂RR activity and C₂H₄ selectivity have been better understood and clarified. Therefore, I think this manuscript can now be considered for publication in Nature Communications.

Reviewer #3 (Remarks to the Author):

In this manuscript, Kim et al made many revisions to the previous version. It is impressive that the authors contributed to this manuscript a lot with noticeable efforts. However, the most pronounced issue is that the authors fail to explain in a simple way the complexity of the reaction system and unclarified process. Besides, the data they provided can not fully prove their claims.

General comments lie in the following parts:

1. As indicated in Fig. R10, the AA could be electrochemically reduced to DHA at -0.36V vs Ag/AgCl in 0.1M KHCO₃, (i.e., + 0.2V vs RHE). However, all the experiments for the electrochemical CO₂ reduction on Cu-based catalysts take place below -0.2V vs RHE. In that way, AA-to-DHA is not reversible. Thus, the real working molecule should be DHA.
2. Following #1, in Fig. R25, DHA-modified Cu NW exhibited better faradaic efficiency (FE) for C₂H₄ production than that from AA-CuNW.
3. From the aspect of the calculated Gibbs energy, the production of CO is much easier than fulfilling C-C coupling. Therefore, how to understand the AA can not only contribute to *CO but also C-C coupling?
4. The authors tried to use in situ IR and Raman to understand the behaviour of AA for CO₂ capture and related interaction. However, the author should also monitor the related blank experiments. Specifically, In the N₂/CO₂ atmosphere, AA/DHA modified Cu NWs. The authors provided the data for the decrease of ·CO₂· with the negative shift of potential (0.2V, 0V vs RHE), but should also present the increase of Cu-C-O.

Author Actions in light of Reviewers Comments

Title: Vitamin C-induced CO₂ capture enables high-rate ethylene production in CO₂ electroreduction

Authors: Jongyoun Kim[†], Taemin Lee[†], Hyun Dong Jung[†], Minkyung Kim, Jungsu Eo, Byeongjae Kang, Hyeonwoo Jung, Jaehyoung Park, Daewon Bae, Yujin Lee, Sojung Park, Wooyul Kim, Seoin Back*, Youngu Lee* and Dae-Hyun Nam*

We sincerely appreciate the reviewers' comments on our manuscript. Followings are the point-by-point response to the comments. We use black font for referee comments and blue font for our response.

Reviewer #1

The authors have made revisions and addressed my comments, and the quality of the work has been highly improved. I recommend the publication.

➤ Response:

We sincerely appreciate your insightful comments and feedback to our work.

Reviewer #2

The authors have made substantial efforts to address the previously raised comments in a satisfactory manner. After the revision, the quality of the manuscript has been greatly improved with the new data and discussions. Particularly, the role and underlying mechanisms of the nanoconfined AA for improving the CO₂RR activity and C₂H₄ selectivity have been better understood and clarified. Therefore, I think this manuscript can now be considered for publication in Nature Communications.

➤ **Response:**

We sincerely appreciate your constructive comments and feedback to our work.

Reviewer #3

In this manuscript, Kim et al made many revisions to the previous version. It is impressive that the authors contributed to this manuscript a lot with noticeable efforts. However, the most pronounced issue is that the authors fail to explain in a simple way the complexity of the reaction system and unclarified process. Besides, the data they provided can not fully prove their claims.

➤ **Response:**

We appreciate your significant concerns about our work. Based on your feedback, we have taken additional studies to address each concern and fully prove our claims. Our finding is that the enol group of ascorbic acid (AA) provides electron/proton and hydrogen bond to CO₂, promoting CO₂-to-*CO conversion and *CO dimerization with reversible AA/dehydroascorbic acid (DHA) redox cycling. We investigated linear sweep voltammetry (LSV) and Fourier transform infrared spectroscopy (FT-IR) to clearly demonstrate the working molecule of cAA-CuNW during CO₂RR. As a result, we confirmed that redox of AA/DHA can be reversible in the potential range of CO₂RR and AA shows the promotional effect on CO₂-to-*CO conversion. We also performed additional *in situ* Raman analysis to support that AA promotes CO₂-to-*CO conversion and *CO dimerization. Detailed point-by-point responses are as follows.

1. As indicated in Fig. R10, the AA could be electrochemically reduced to DHA at -0.36 V vs Ag/AgCl in 0.1 M KHCO_3 , (i.e., $+ 0.2$ V vs RHE). However, all the experiments for the electrochemical CO_2 reduction on Cu-based catalysts take place below -0.2 V vs RHE. In that way, AA-to-DHA is not reversible. Thus, the real working molecule should be DHA.

➤ **Response:**

Since the carbonyl group in DHA does not donate electrons and protons to CO_2 , DHA cannot be expected to promote CO_2 -to- $\ast\text{CO}$ conversion and $\ast\text{CO}$ dimerization. In contrast, previous reports showed that AA can be used as an electron donor to CO_2 in liquid-gas interface (**Fig. R1**).¹⁻³ Pastero et al. demonstrated a bubble-drop system for the mineral capture of CO_2 by using AA as a sacrificial reductant.^{1,2} They used Ca ascorbate (CaAsc) solution to promote precipitation of Ca oxalate (CaC_2O_4) by transferring electrons from AA to CO_2 , achieving CO_2 capture into a stable, safe, and insoluble salt. Cardoso et al. also reported that self-oxidation of AA increased the availability of electrons in the electrolyte, contributed to the photoelectrocatalytic reduction of CO_2 on $\text{TiO}_2/\text{ZIF-8}$ to MeOH.³ Therefore, we think that AA should be considered as the major molecule contributing to CO_2RR .

Our findings on the introduction of AA in electrochemical CO_2RR are that nanoconfined AA on GQDs is regenerated *via* CO_2RR under reduction potential, ensuring persistent promotion of CO_2 -to- $\ast\text{CO}$ conversion and $\ast\text{CO}$ dimerization. To investigate redox behavior of AA/DHA in the potential range of CO_2RR , we measured LSV of nanoconfined AA on GQDs before and after CO_2RR (**Fig. R2**). LSV was measured with 0.1 M KHCO_3 electrolyte to minimize the changes in the curve due to pH drop from bicarbonate formation. We observed the oxidation of AA at 0.57 V (vs RHE, non-iR corrected) before CO_2RR (**Fig. R2a**). However, the oxidation peak disappeared for the 2nd scan, indicating that most AA was oxidized to DHA during LSV measurement.

After confirming the decrease of AA oxidation in the LSV of nanoconfined AA on GQDs before CO_2RR (red), the CO_2RR of nanoconfined AA on GQDs was conducted at a constant potential of -1.8 V (vs RHE, non-iR corrected) for 24 h (**Fig. R2b**). In the LSV curve after CO_2RR (blue), we observed the oxidation of AA at 0.62 V (vs RHE, non-iR corrected), which is shifted due to the additional energy required to break the interaction

between CO₂ and AA (**Fig. R2b**).^{4,5} The current density of peak for AA oxidation was higher in the nanoconfined AA after 24 h CO₂RR compared to that before CO₂RR (**Fig. R2b**). This indicates that DHA was reduced to AA during electrochemical CO₂RR, regenerating AA to continuously donate electron and proton to CO₂.

In addition, we further analyzed the chemical state of cAA-CuNW by FT-IR to confirm the presence of AA or DHA after CO₂RR (**Fig. R3**). FT-IR spectra of cAA-CuNW after CO₂RR showed strong C=C peak corresponding to AA, while no C=O peak (1,770 cm⁻¹) corresponding to DHA was observed.⁶ We believe that the dominant AA detection in LSV and FT-IR is due to the reduction of DHA to AA and excessive AA which does not participate in CO₂RR. Based on these results, we propose that the major molecule for CO₂RR promotion is AA, which exists through reversible redox in the potential range of CO₂RR.

➤ **Modification:**

(Manuscript page 9 line 207-213) We add the explanation about the result of LSV analysis.

(Supplementary information page 19) Supplementary Fig. 16 has been redrawn as Fig. R4 to highlight the AA/DHA redox.

(Supplementary information page 21) Fig. R2 has been added as Supplementary Fig. 18.

L. Pastero et al., *Sci. Total Environ.* **666**, 1232-1244 (2019).

J. C. Cardoso et al., *J. Electroanal. Chem.* **856**, 113384 (2020).

Fig. R1 | Previous reports on the electron donating properties of AA for CO₂ capture and reduction. **a** Bubble-drop system for the mineral capture of CO₂ using Ca ascorbate (CaAsc) as a sacrificial reductant. **b** Reaction of CaAsc with CO₂ to synthesize Ca oxalate (CaC₂O₄) precipitate (top) and the amount of captured CO₂ in 1 M CaAsc solution with and without bubble-drop system (bottom). Reproduced from L. Pastero et al., *Sci. Total Environ.* **666**, 1232-1244 (2019).¹ **c** Comparison of photoelectrocatalysis (PEC), photocatalysis (PC) and photolysis (PT) processes on TiO₂/ZIF-8 with AA (left) and methanol production of PEC on TiO₂/ZIF-8 with and without AA dissolved in electrolyte (right). **d** Methanol (MeOH) production by PEC process on TiO₂/ZIF-8 with self-oxidation of AA. Reproduced from J. C. Cardoso et al., *J. Electroanal. Chem.* **856**, 113384 (2020).³

Fig. R2 | LSV of nanoconfined AA on GQDs before and after CO₂RR. LSV of nanoconfined AA on GQDs (a) before and (b) after CO₂RR in 0.1 M KHCO₃ electrolyte. c Flow chart to investigate the redox behavior of AA/DHA in the potential range of LSV and CO₂RR. Nanoconfined AA on GQDs was prepared on glassy carbon electrode (GCE) and CO₂RR was conducted at a constant potential of -1.8 V (vs RHE, non-iR corrected) in 0.1 M KHCO₃ electrolyte.

Fig. R3 | FT-IR spectra of DHA and cAA-CuNW after CO_2RR stability test. The peak at $1,770 \text{ cm}^{-1}$ corresponds to the C=O stretch mode of DHA.⁶

Fig. R4 | Redox behavior of AA and nanoconfined AA on GQDs under N₂ and CO₂ gas. a CV plot of GCE with AA dissolved in 0.1 M KHCO₃ electrolyte. **b** CV plot of GCE coated with nanoconfined AA on GQDs in 0.1 M KHCO₃ electrolyte. 4th cycle of the CV plot of GCE coated with nanoconfined AA on GQDs under (c) N₂ and (d) CO₂ gas.

2. Following #1, in Fig. R25, DHA-modified Cu NW exhibited better faradaic efficiency (FE) for C₂H₄ production than that from AA-CuNW.

➤ **Response:**

We fabricated DHA-CuNW by coating CuNW with DHA and there was no GQDs in DHA-CuNW. Note that DHA is not reduced to AA in DHA-CuNW during CO₂RR due to the absence of a nanoconfinement effect by GQDs. We further analyzed the CO₂RR performance of DHA-CuNW up to -1.47 V (vs RHE) to clearly confirm the effect of DHA on CO₂RR (**Fig. R5**). DHA-CuNW did not exhibit significant CO production with the CO FE of 4.5% at -0.87 V (vs RHE) (**Fig. R5a**), while AA-CuNW and cAA-CuNW presented dramatically elevated CO FE of 17.0 and 36.7% at -0.81 V (vs RHE) (**Fig. 3c, d**). As the potential increased to -1.47 V (vs RHE), H₂ FE of DHA-CuNW increased from 5.8% to 23.2% and C₂H₄ FE decreased from 58.3% to 34.5% (**Fig. R5a**), which is similar to the CO₂RR of p-CuNW (**Fig. 3a**).

We think that the enhancement of C₂H₄ FE in DHA-CuNW at -0.87 V (vs RHE) compared to that of p-CuNW at -0.85 V (vs RHE) might be caused by the existence of organic capping agents in DHA-CuNW which can stabilize Cu active materials (**Fig. R5b**).^{7,8} Although the C₂H₄ FE has been slightly increased in low overpotential range in DHA-CuNW, gradual decrease of C₂H₄ FE and increase of H₂ FE were observed as the potential increased to -1.47 V (vs RHE). Therefore, we verified that the effect of DHA does not significantly contribute to high-rate C₂H₄ production. DHA does not promote CO₂-to-*CO formation due to the absence of the enol group, which donates electrons and protons to CO₂.

➤ **Modification:**

(**Supplementary information page 31**) We add the CO₂RR results of DHA-CuNW at extended potential range (Fig. R5a) in Supplementary Fig. 28.

Fig. R5 | Effect of DHA on electrochemical CO₂RR performance of CuNW. a Gaseous product FEs and total current densities of DHA-CuNW up to -1.47 V (vs RHE). **b** Comparison of the gaseous product selectivity of p-CuNW, DHA-CuNW, and AA-CuNW.

3. From the aspect of the calculated Gibbs energy, the production of CO is much easier than fulfilling C-C coupling. Therefore, how to understand the AA can not only contribute to *CO but also C-C coupling?

➤ **Response:**

We thank the reviewer for pointing this out. As we can see from the DFT results, *CO formation is more favorable than C-C coupling, both in the presence and absence of AA. We note that AA's presence further enhances the favorability of *CO formation, leading to more efficient CO production. This aligns with the experimental results shown in **Fig. 3a–d**, where cAA-CuNW produced more CO than other CuNWs in low overpotential regions due to higher *CO coverage. In addition, high *CO coverage can promote the coupling of *CO; the partial current density of C₂H₄ is proportional to the square of *CO coverage.⁹ In terms of C-C coupling, AA's presence lowers the activation barrier for the coupling of *CO, thus accelerating C₂H₄ production.

4. The authors tried to use *in situ* IR and Raman to understand the behaviour of AA for CO₂ capture and related interaction. However, the author should also monitor the related blank experiments. Specifically, In the N₂/CO₂ atmosphere, AA/DHA modified Cu NWs. The authors provided the data for the decrease of *CO₂ with the negative shift of potential (0.2 V, 0 V vs RHE), but should also present the increase of Cu-C-O.

➤ **Response:**

We performed further analysis in *in situ* Raman spectroscopy to fully prove that AA promotes CO₂-to-*CO conversion and *CO dimerization. We confirmed the Cu-CO binding peaks (200-400 cm⁻¹) for p-CuNW and cAA-CuNW at the potential from 0.2 to –0.2 V (vs RHE, non-iR corrected) (**Fig. R6**). The intensity of Cu-CO stretching peak (300-400 cm⁻¹) for p-CuNW increased from –0.1 V (vs RHE, non-iR corrected), accompanied by the decrease of *CO₂ peak intensity. However, Cu-CO stretching peak for cAA-CuNW emerged from 0.1 V (vs RHE, non-iR corrected) and showed higher intensity than p-CuNW due to the enhanced CO₂-to-*CO conversion.

In addition, we analyzed *in situ* Raman spectroscopy of p-CuNW and cAA-CuNW under CO₂+Ar mixed gas at potentials ranging from –0.4 to –0.8 V (vs RHE, non-iR corrected) to further prove the promotion effect of nanoconfined AA on GQDs (**Fig. R7**). The intensity of Cu–CO binding peaks at 200-400 cm⁻¹ is higher for cAA-CuNW than that of p-CuNW, indicating that cAA-CuNW shows high *CO coverage even at low CO₂ concentration. In the C=O stretching region, Raman spectra under CO₂+Ar mixed gas also showed similar trends. CO_{atop} (1,950-2,000 cm⁻¹) and CO_{bridge} (2,050-2,100 cm⁻¹) were also formed in cAA-CuNW under CO₂+Ar mixed gas, whereas the CO_{bridge} peak almost disappeared as the reductive potential increased in p-CuNW. This corresponds to the enhanced CO₂RR performance of cAA-CuNW at low CO₂ concentration compared to that of p-CuNW (**Fig. 3g–h** and **Supplementary Fig. 35**), proving the promotion of CO₂-to-*CO conversion and *CO dimerization in the presence of nanoconfined AA on GQDs.

We also measured *in situ* Raman spectroscopy of CuNWs under N₂ gas to support the monitoring of *CO bindings (**Fig. R8** and **R9**). All CuNWs under N₂ gas showed no peaks at 200-400 cm⁻¹, suggesting that Cu-CO binding peaks under CO₂ gas were originated from gaseous CO₂. In addition, we measured *in situ* Raman spectroscopy of

DHA-CuNW to verify the promotion of CO₂-to-*CO conversion by DHA. However, *in situ* Raman spectroscopy of DHA-CuNW showed lower intensity of Cu–CO binding peaks than that of cAA-CuNW (**Fig. R9b** and **Fig. 4g**), which corresponds to the poor CO production in the low potential range. In addition, disappearance of CO_{bridge} was observed as the reductive potential increased in DHA-CuNW (**Fig. R9c**). Therefore, we confirmed that DHA does not promote CO₂-to-*CO conversion without electrochemical reduction of DHA to AA.

➤ **Modification:**

(Manuscript page 14 line 326-328) We add explanation about the analysis in *in situ* Raman spectroscopy of CuNWs under N₂ gas.

(Manuscript page 14-15 line 331 and 344) We add *in situ* Raman spectroscopy of DHA-CuNW to compare with other CuNWs.

(Manuscript page 15 line 347-349) We add explanation about the analysis in *in situ* Raman spectroscopy of CuNWs under CO₂+Ar mixed gas.

(Manuscript page 39 line 837-838) We revise the figure caption for better clarity.

(Supplementary information page 43) Fig. R8a, c, e, and g has been added as Supplementary Fig. 40.

(Supplementary information page 44) Fig. R9 has been added as Supplementary Fig. 41.

(Supplementary information page 45) Fig. R6 has been added as Supplementary Fig. 42. Original graphs have been added as Supplementary Fig. 42b and d.

(Supplementary information page 46) Fig. R7 has been added as Supplementary Fig. 43.

Fig. R6 | Monitoring ^{*}CO₂⁻ on CuNWs during CO₂RR. *In situ* Raman spectra of (a, b) p-CuNW and (c, d) cAA-CuNW obtained during CO₂RR according to the applied potentials in the region of 200-700 cm⁻¹ (top) and 1,400-1,700 cm⁻¹ (bottom).

Fig. R7 | Real-time observation of *CO bindings from CuNWs under CO₂+Ar mixed gas. *In situ* Raman spectra of (a, b) p-CuNW, and (c, d) cAA-CuNW obtained during CO₂RR according to the applied potentials in the region of 200-700 cm⁻¹ (top) and 1,800-2,400 cm⁻¹ (bottom).

Fig. R8 | Real-time observation of *CO bindings from CuNWs under N_2 and CO_2 gas. *In situ* Raman spectra of (a, b) p-CuNW, (c, d) G-CuNW, (e, f) AA-CuNW, and (g, h) cAA-CuNW obtained in the region of 200-700 cm^{-1} according to the applied potentials under N_2 (top) and CO_2 gas (bottom).

Fig. R9 | Real-time observation of *CO bindings from DHA-CuNW. *In situ* Raman spectra of DHA-CuNW obtained in the region of 200-700 cm⁻¹ according to the applied potentials under (a) N₂ and (b) CO₂ gas. c *In situ* Raman spectra of DHA-CuNW obtained in the region of 1,800-2,400 cm⁻¹ according to the applied potentials under CO₂ gas.

References

1. Pastero, L. et al. CO₂ capture and sequestration in stable Ca-oxalate, via Ca-ascorbate promoted green reaction. *Sci. Total. Environ.* **666**, 1232-1244 (2019).
2. Pastero, L., Marengo, A., Boero, R. & Pavese, A. Non-conventional CO₂ sequestration via vitamin C promoted green reaction: Yield evaluation. *J. CO₂ Util.* **44**, 101420 (2021).
3. Cardoso, J. C. et al. The effective role of ascorbic acid in the photoelectrocatalytic reduction of CO₂ preconcentrated on TiO₂ nanotubes modified by ZIF-8. *J. Electroanal. Chem.* **856**, 113384 (2020).
4. Choi, G. H. et al. Electrochemical direct CO₂ capture technology using redox-active organic molecules to achieve carbon-neutrality. *Nano Energy* **112**, 108512 (2023).
5. Iijima, G. et al. Mechanism of CO₂ capture and release on redox-active organic electrodes. *Energy Fuels* **37**, 2164-2177 (2023).
6. Gupta, H., Paul, P., Kumar, N., Baxi, S. & Das, D. P. One pot synthesis of water-dispersible dehydroascorbic acid coated Fe₃O₄ nanoparticles under atmospheric air: Blood cell compatibility and enhanced magnetic resonance imaging. *J. Colloid Interface Sci.* **430**, 221-228 (2014).
7. Fan, Q. et al. Manipulating Cu nanoparticle surface oxidation states tunes catalytic selectivity toward CH₄ or C₂⁺ products in CO₂ electroreduction. *Adv. Energy Mater.* **11**, 2101424 (2021).
8. Liu, H. et al. Polydopamine functionalized Cu nanowires for enhanced CO₂ electroreduction towards methane. *ChemElectroChem* **5**, 3991-3999 (2018).
9. Gao, D. et al. Activity and selectivity control in CO₂ electroreduction to multicarbon

products over CuO_x catalysts via electrolyte design. *ACS Catal.* **8**, 10012-10020 (2018).

REVIEWERS' COMMENTS

Reviewer #3 (Remarks to the Author):

The authors have made revisions and addressed my comments, and the quality of the work has been highly improved. I recommend the publication.